# Society of Mind Meets Real-Time Strategy: A Hierarchical Multi-Agent Framework for Strategic Reasoning

**Daechul Ahn,**[*] **San Kim**[*]**, Jonghyun Choi**[†]
Seoul National University
{daechulahn,00sankim,jonghyunchoi}@snu.ac.kr

## Abstract

Large Language Models (LLMs) have recently demonstrated impressive action sequence prediction capabilities but often struggle with dynamic, long-horizon tasks such as real-time strategic games. In a game such as StarCraft II (SC2), agents need to manage resource constraints and adapt to evolving battlefield situations in a partially observable environment. This often overwhelms exisiting LLM-based approaches. To address these challenges, we propose a hierarchical multi-agent framework that employs specialized imitation learning agents under a meta-controller called *Strategic Planner* (SP). By expert demonstrations, each specialized agent learns a distinctive strategy, such as aerial support or defensive maneuvers, and produces coherent, structured multistep action sequences. The SP then orchestrates these proposals into a single, environmentally adaptive plan that ensures local decisions aligning with long-term strategies. We call this HIMA (**Hi**erarchical **I**mitation **M**ulti-**A**gent). We also present TEXTSCII-ALL, a comprehensive SC2 testbed that encompasses all race match combinations in SC2. Our empirical results show that HIMA outperforms state of the arts in strategic clarity, adaptability, and computational efficiency, underscoring the potential of combining specialized imitation modules with meta-level orchestration to develop more robust, general-purpose AI agents. We release our code, model and datasets at ⚙ https://github.com/snumprlab/hima.

## 1 Introduction

Large Language Models (LLMs) have achieved remarkable success in various tasks (Brown et al., 2020; Raffel et al., 2020; Ouyang et al., 2022; Touvron et al., 2023) but its ability to understand complex, ever-changing environments to reason a sequence of actions remains challenging (Brohan et al., 2022; Driess et al., 2023; Fan et al., 2022). Generating proper action sequences in the changing environment requires *strategic reasoning* to address uncertainty of the changing environment (Gandhi et al., 2023). We argue that real-time strategy (RTS) games exemplify these challenges. In StarCraft II (SC2) (Blizzard Entertainment, 2010), for instance, players must gather resources, build infrastructure, manage expansions, and produce units simultaneously in real time, while operating under partial observability of the battlefield. This interplay of short-term tactics (*e.g.*, efficient early builds) and non-trivial objectives (*e.g.*, achieving aerial dominance over the battlefield) makes SC2 a *well-suited* domain to test strategic reasoning (Vinyals et al., 2019).

Recently, Ma et al. (2024) propose TextStarCraft II, a text-based evaluation environment for SC2 where game states and actions are processed as text, featuring multiple controllable units, diverse resource requirements, and restricted visibility in an evolving battlefield (see details in Appendix Sec. B). Existing LLM-based approaches (Ma et al., 2024; Shao et al., 2024; Wu & Hu, 2025; Li et al., 2025b) can interpret game states to predict a sequence of actions, but they often struggle to maintain efficient resource management and coherent build orders,

---

[*]These authors contributed equally.
[†]JC is with ECE, ASRI and IPAI in SNU and a corresponding author.

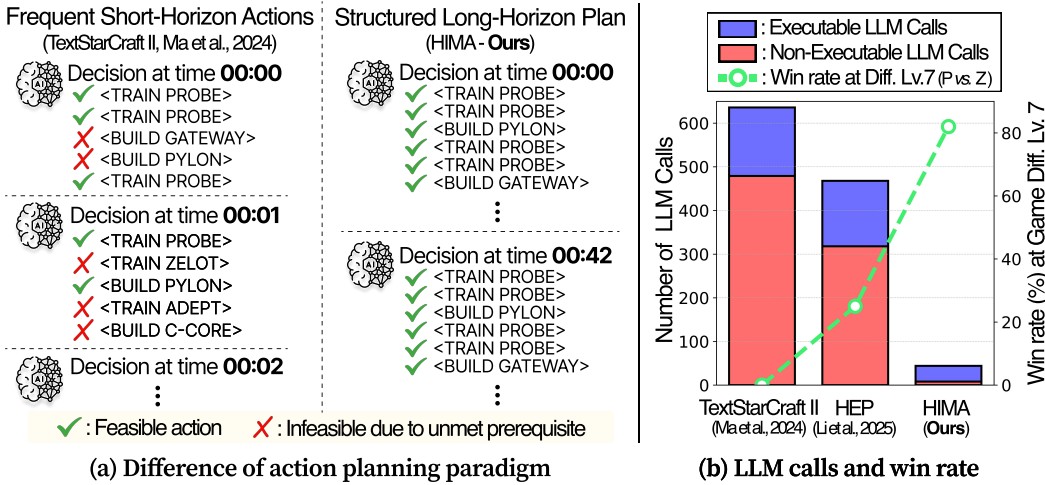

Figure 1: **Comparison of short-horizon (Ma et al., 2024) *vs.* proposed long-horizon (HIMA) planning in SC2.** Fewer LLM calls, more coherent build orders, and higher win rates in HIMA. (a) Existing methods frequently produce one-step actions at short intervals (*e.g.*, 00:00, 00:01, 00:02, . . . ), causing repeated invalid actions (X) due to unmet prerequisites. In contrast, HIMA produces structured multi-step plans with fewer queries (*e.g.*, 00:00, 00:42), respecting build orders and maintaining coherent strategy. (b) Quantitative results showing LLM calls over average of 20 minutes of gameplay (blue: executable; red: non-executable) and win rate against level 7 AI (Protoss *vs.* Zerg; green line). HIMA requires fewer LLM calls while achieving higher win rates, demonstrating superior efficiency and effectiveness.

*e.g.*, producing actions that ignore required prerequisites or repeatedly producing redundant commands, wasting resources, and disrupting build orders, as shown in Fig. 1.

One way to mitigate these issues is to use imitation learning (IL) (Abbeel & Ng, 2004; Torabi et al., 2018; Urakami et al., 2024; Wu et al., 2025), which uses human demonstrations to capture the core patterns of expert play. Although IL can impart valuable strategy to a model, a purely imitation-driven approach may fail when faced with novel or evolving battlefield scenarios in which the opponent's capabilities differ from those seen in the training data. Furthermore, in RTS games such as SC2, multiple paths to victory exist — such as early aggressive rushes, aerial dominance, or balanced resource management — require agents to adapt to diverse strategic routes in each situation. This breadth of strategies, each with distinct requirements and tactics, often poses challenges for a single monolithic model.

Another way is to use reinforcement learning (RL) (Vinyals et al., 2019). However, it requires much computation for many trials and errors, which may not justify the cost versus efficacy trade-off, and a well-designed reward function, which is often not trivial.

To address these limitations in a cost-effective manner, we draw inspiration from the 'society of mind' principle (Minsky, 1986) and propose a multi-agent framework that coordinates specialized imitation learning agents under a central meta-controller (called the *Strategic Planner* (SP)), which we collectively refer to as HIMA (**H**ierarchical **I**mitation **M**ulti-**A**gent), as shown in Fig. 2. Each specialized agent generates longer-horizon *structured action sequences* (Sec. 3.1), ensuring that procedural prerequisites are fulfilled and reducing invalid or redundant commands (Fig. 1). Before committing to a final decision, the SP then synthesizes these proposals through an *environment-aware action orchestration*, using proposed 'temporal Chain-of-Thought (t-CoT)' reasoning. It induces the SP to align the final decision with immediate, short-term, and long-term objectives (Sec. 3.2). In particular, the SP continuously monitors the changing battlefield conditions, adapting its strategic decision based on changing environments before finalizing action plans. Using longer-horizon structured action sequences from imitation agents, the SP requires significantly fewer adjustments than the prior arts that frequently query LLMs at every time step, as illustrated in Fig. 1.

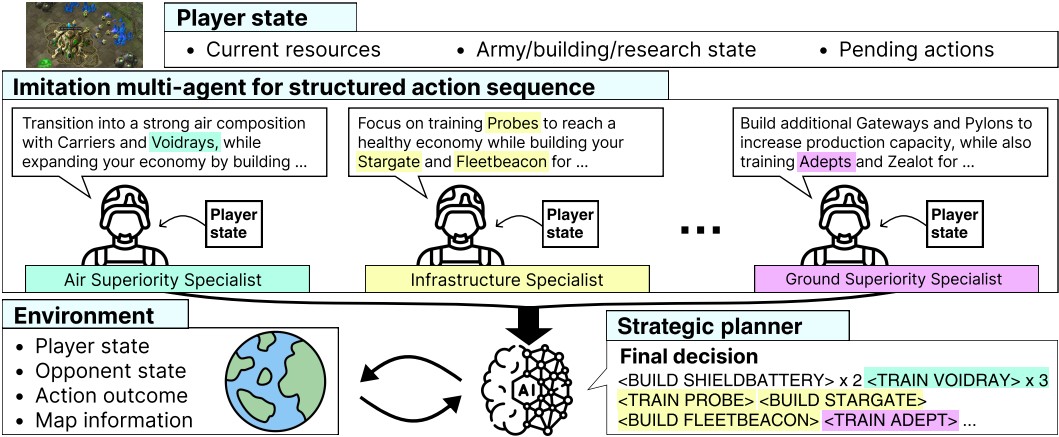

Figure 2: **Overview of the proposed hierarchical imitation multi-agent (HIMA) framework**. Each specialized imitation agent (*e.g.*, Air Superiority, Infrastructure, Ground Superiority) receives up-to-date *player state* information (resources, units, buildings) and produces a multi-step action plan along with a rationale explaining its strategic intent. A high-level meta-controller (*i.e.*, Strategic Planner) then merges these agent-level proposals into a single cohesive decision, factoring in broader environmental contexts (opponent state, action outcomes, battle progress). This hierarchical structure enables long-horizon planning, adaptive coordination among specialized agents, and real-time responsiveness to evolving battlefield.

| SC2 Evaluation Environment | Player Race(s) | Opponent Race(s) | #Matchups |
|---|---|---|---|
| TextStarCraft II (Ma et al., 2024) | Protoss | Zerg | 1 |
| SwarmBrain (Shao et al., 2024) | Zerg | Terran | 1 |
| **TextSCII-All (Ours)** | Protoss, Zerg, Terran | Protoss, Zerg, Terran | 9 |

Table 1: **Comparison of text-based SC2 evaluation environment.** Unlike previous work with a single player race and limited enemy matchups (one each), our TextSCII-All covers all three races on both sides, resulting in a total of nine possible match-ups.

In addition, to evaluate SC2 comprehensively, we present an expanded SC2 evaluation environment that encompasses all three races (Protoss, Terran and Zerg) and their nine possible match-ups, as shown in Tab. 1, which we call TextSCII-All. Beyond single match-up studies in the literature (Ma et al., 2024), new environment exhibits a comprehensive assessment of matches in all race combinations. We empirically validate methods in this comprehensive environment, demonstrating that HIMA outperforms state-of-the-arts (SoTAs) in win rate against built-in AI, head-to-head matches, and computational efficiency.

We summarize contributions as follows: (1) we propose a hierarchical imitation multi-agent framework that enables efficient and effective long-term strategic planning in SC2; (2) we expand the SC2 evaluation environment to include all nine race matchups, offering a more comprehensive benchmarking testbed; and (3) we demonstrate the effectiveness of our proposed HIMA against both built-in AI opponents and direct match-ups with SoTAs.

## 2 Related Work

**Interactive environment.** Real-time strategic decision-making environments have progressively increased in complexity. Tasks have expanded within physics-based simulations and gaming platforms, evolving from simpler frameworks like MuJoCo (Todorov et al., 2012) and ALE (Bellemare et al., 2013) to sophisticated open-ended systems including MineCraft (Wang et al., 2024; Li et al., 2025a), PokeLLMon (Hu et al., 2024), ALFWorld (Shridhar et al., 2021), and ScienceWorld (Wang et al., 2022), among others (Albrecht et al., 2022; Qi et al., 2024). These systems often involve partial observability and tasks mirroring real-world scenarios.

Recent LLM-based SC2 testbeds, including TextStarCraft II (Ma et al., 2024) and Swarm-Brain (Shao et al., 2024), present formidable obstacles for language models and autonomous agents. They demand intensive operational control, navigation through expansive action spaces, processing of complex state representations, and long-range planning over extended gameplay sequences, underscoring modern interactive environments' heightened difficulty.

**Multi-agent interaction.** The *Society of Mind* theory proposed by Minsky (Minsky, 1986) posits that intelligence materializes through interactions between specialized agents—a foundation in multi-agent systems (MAS). Within MAS frameworks, autonomous agents engage in either collaborative or competitive behaviors to facilitate collective reasoning across numerous domains (Zhang et al., 2019; Chen et al., 2024a; Park et al., 2023; Chen et al., 2023; Li et al., 2023). The advent of large language models (LLMs) has catalyzed the development of advanced multi-agent architectures, encompassing methodologies such as Voting (Wang et al., 2023), Debate (Du et al., 2023), Reconcile (Chen et al., 2024b), Role-playing (Tseng et al., 2024; Long et al., 2024), and comprehensive societal simulations (Paquette et al., 2019; Qi et al., 2024). Here, we present HIMA, a multi-agent framework that employs imitation-based interactions coordinated by a sophisticated meta-controller for real-time strategic reasoning task such as SC2. This meta-controller adaptively synthesizes environmental contexts with agent-specific objectives, enabling cohesive inter-agent communication that markedly improves strategic coordination and system efficacy.

## 3 HIMA: A Hierarchical Multi-Agent Imitation Framework

To address the limitations of existing LLM-based methods such as producing short-horizon actions and invalid build orders, we propose HIMA, a hierarchical multi-agent architecture (Fig. 2) that deploys specialized imitation agents trained on human demonstration data for generating diverse structured action sequences (Sec. 3.1), coordinated by a Strategic Planner (SP) that integrates these proposals and aligns them with overarching objectives while considering broader environmental contexts (Sec. 3.2).

### 3.1 Learning Multi-Agent for Structured Action Sequence

In SC2, the number of actions executed per timestep increases as the match progresses, as observed in the human demonstration (Appendix Sec. H). The early stages are constrained by limited resources and procedural prerequisites, while the later stages enable a diverse action space and require more actions per time step to manage the increasing complexity of units, buildings, and resources in an evolving battlefield. Existing approaches that generate a fixed number of actions per timestep (Ma et al., 2024; Li et al., 2025b; Wu & Hu, 2025) often fail to keep pace with this evolving "tempo," producing frequently fragmented strategies.

**Structured action sequences with rationales.** To address the strategy fragmentation, we take inspiration from *human gameplay*, where players naturally vary their frequency of action as the match progresses, taking into account both immediate strategic conditions and future objectives. Specifically, rather than producing fixed action counts $A_t$ per decision point, we design our agents to generate *structured action sequences* that span a time window $\Delta$, denoted as $A_{t:t+\Delta}$. This enables variable action density within each window, reflecting gameplay patterns where later phases require more actions in the same timeframe. To guide these variable-density action sequences, we also append a *Tactical Rationale* (TR) explaining why each action set suits its particular game phase and context. This explicit reasoning improves decision quality by providing strategic context (see Appendix Sec. J for empirical benefits).

We implement this idea in three steps as follows (illustrated in Fig. 3-(a)). First, we extract state-action pairs $\{S_t, A_t\}$ from the SC2EGSet (Białecki et al., 2023), a dataset containing professional SC2 replays annotated with detailed game states and corresponding player actions (see Fig. 3 (left) and Appendix E.1). Then, we prompt an LLM (GPT-4o-mini (OpenAI, 2024)) to generate TR for action sequences $A_{t:t+\Delta}$. Finally, we convert the results into the instruction tuning format for supervised fine tuning (SFT). We provide detailed construction

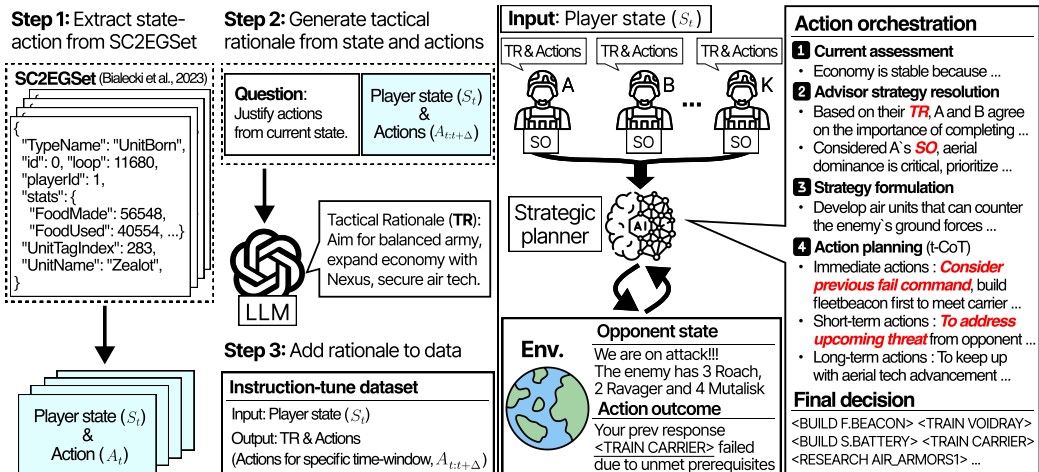

Figure 3: **Dataset construction pipeline for generating structured action sequence and environment-aware action orchestration in HIMA.** (a) We extract state-action pairs $\{S_t, A_t\}$ from SC2EGSet (Białecki et al., 2023) and prompt an LLM to generate a *Tactical Rationale* (TR) for each multi-step action sequence, $A_{t:t+\Delta}$. This TR is appended to form an instruction-tuning dataset, capturing *what* actions occur and *why* they're chosen. (b) In deployment, multiple specialized agents (A, B, . . . , K) produce pairs of *Strategic Objective* (SO) and TR, which the strategic planner reconciles based on current player and opponent states. This environment-aware process ensures adaptive short-term responses—*e.g.*, mitigating failed commands—and consistent long-term planning (*e.g.*, securing air superiority) by integrating rationales and outcome feedback from ongoing game events.

procedures in Appendix Sec. E.1. This approach mirrors human gameplay, where the frequency of action increases with the complexity of the game.

**Multi-agent specialization.** Even with structured action sequences, covering the vast strategic space of SC2 remains a challenge for a single model. For instance, one strategy might focus on quick air dominance (using mass flying units), while another aims at a well-fortified ground army. Thus, we adopt a multi-agent approach (Wang et al., 2023; Du et al., 2023; Chen et al., 2024b; Long et al., 2024), where each agent specializes in a particular 'strategic persona'. Specifically, we leverage the instruction-tuned data and group the agents by *unit compositions* (*e.g.*, ground-heavy vs. air-heavy armies) which is a crucial factor influencing build orders and combat styles in SC2 using *k*-means clustering. Since similar compositions often indicate consistent strategic patterns (*e.g.*, mass air vs. strong ground), each cluster produces a specialized agent. Combining these agents covers a wider range of strategies than a single model. Subsequently, a meta-controller (Sec. 3.2) merges their action sequences into a unified one that adapts to real-time developments.

## 3.2 Strategic Planner

In HIMA, the SP acts as a meta-controller that fuses structured action sequences from specialized imitation agents into a coherent long-range plan, as illustrated in Fig. 3-(b).

**Environment-aware action orchestration.** We plan actions in four stages as follows.

1. **Current assessment:** The SP examines the latest game state (*e.g.*, resources, unit compositions, visible enemy units) to establish current context.

2. **Advisor strategy resolution:** The SP then applies the Nominal Group Technique (NGT) (Delbecq & de Ven, 1971; Long et al., 2024), a structured decision-making method, to systemically analyze multiple agents' strategic proposals, as shown in Fig. 3-(b).

Specifically, the SP applies a structured four-step process: (i) identifying agreed and conflicted viewpoints among agents; (ii) resolving conflicts by evaluating each agent's rationale; (iii) considering isolated but insightful viewpoints; and (iv) synthesizing these inputs into a cohesive final strategy. In particular, the SP resolves the strategies by explicitly considering the intentions behind each action proposed by the agents. To achieve this, the SP evaluates each agent's *TR* (justification for the proposed actions) along with its corresponding *Strategic Objective* (SO), a high-level strategic goal derived from expert gameplay analysis (see Appendix Sec. K for details and Fig. 10 for examples).

3. **Strategy formulation:** The SP balances each agent's logic, goals, and battlefield constraints to produce a consolidated action plan.

4. **Temporal Chain-of-Thought:** Finally, the SP breaks down the chosen strategy into *immediate actions* (urgent responses), *short-term actions* (near-term objectives), and *long-term actions* (larger strategic goals) before generating a final decision. We refer to this systematic alignment of different time horizons as a *temporal Chain-of-Thought (t-CoT)* as it connects immediate, short-term, and long-term goals into a step-by-step reasoning chain that reflects how players plan over time.

By the four stages, the SP effectively orchestrates the specialized agents' actions, maintaining coherent decision-making in an evolving game state (see more detailed prompts in Figures 14 and 15). Furthermore, since each imitation agent produces a structured action sequence over a fixed time window $A_{t:t+\Delta}$, the SP requires significantly fewer plan updates compared to previous methods (Ma et al., 2024; Wu & Hu, 2025; Li et al., 2025b) that query LLMs at frequent fixed intervals, substantially reducing LLM call overhead while maintaining strategic depth, as illustrated in Fig. 1-(b).

**Feedback system for an immediate adaptation.** Although long-term planning through structured action sequence is beneficial to efficiently building coherent strategies, it can limit responsiveness if the state of the game changes unexpectedly. To address this, the SP employs a *feedback system* that reexamines the current plan after critical events. If a sequence fails to meet its objective, the SP records the failure and refines subsequent prompts to prevent repeating similar errors, as illustrated in Fig. 3-(b). Likewise, if the agent detects an unexpectedly large enemy force (*i.e.*, when the number of enemy units exceeds a threshold, $\tau$=10, which is heuristically determined), the SP discards the current plan and requests a new action sequence starting from imitation agents. This mechanism blends proactive long-range planning with the real-time adaptability needed to handle sudden changes in RTS gameplay. More comprehensive scenarios demonstrating the feedback system's operation are available in Appendix Sec. O.

By integrating specialized agents' proposals through environment-aware action orchestration and adapting through the feedback system, the SP generates a strategy that responds to changing battlefield conditions.

## 4 Experiments

### 4.1 Experimental Setup

**Training and inference.** We primarily use Qwen-2 1.5B (Yang et al., 2024) as our base imitation learning agent, with additional open-source models described in Appendix Sec. L. For SFT, we extract 39.9k, 39.8k, and 30.8k training instances for Protoss, Zerg, and Terran from human demonstrations (Białecki et al., 2023), converting them to instruction-tuning format. We set $\Delta$ as 3 minutes for structured action sequence based on the experiments (Fig. 4-(c)) and $k = 3$ groups per race validated through empirical testing (Appendix Sec. G) Details about instruction-tuning dataset construction are in Appendix Sec. E. During inference, we primarily employ GPT-4o-mini (OpenAI, 2024) as the SP, with additional evaluations on other open-source and closed-source models described in Appendix Sec. M. Unless otherwise specified, we measure win-rates over 50 randomly sampled matches.

| Method | Player vs. Opponent | Win Rate (%) at Game Difficulty | | | | | | |
|---|---|---|---|---|---|---|---|---|
| | | Lv.4 | Lv.5 | Lv.6 | Lv.7 | Lv.8 | Lv.9 | Lv.10 |
| TextStarCraft (Ma et al., 2024) | Protoss vs. Zerg | 84 | 55 | 8 | – | – | – | – |
| SwarmBrain (Shao et al., 2024) | Zerg vs. Terran | 100 | 76 | – | – | – | – | – |
| EpicStar (Wu & Hu, 2025) | Protoss vs. Zerg | – | 67 | 30 | – | – | – | – |
| HEP (Li et al., 2025b) | Protoss vs. Zerg | 100 | 75 | 75 | 25 | – | – | – |
| **HIMA (Ours)** | Protoss vs. Protoss | 90 | 80 | 68 | 48 | 42 | 14 | 4 |
| | Protoss vs. Terran | 94 | 86 | 70 | 68 | 48 | 24 | 8 |
| | Protoss vs. Zerg | 100 | 92 | 84 | 82 | 68 | 20 | 16 |
| | Zerg vs. Protoss | 70 | 48 | 8 | 0 | 0 | 0 | 0 |
| | Zerg vs. Terran | 100 | 84 | 12 | 8 | 0 | 0 | 0 |
| | Zerg vs. Zerg | 72 | 50 | 12 | 14 | 0 | 0 | 0 |
| | Terran vs. Protoss | 58 | 28 | 16 | 10 | 0 | 0 | 0 |
| | Terran vs. Terran | 62 | 4 | 0 | 0 | 0 | 0 | 0 |
| | Terran vs. Zerg | 60 | 32 | 6 | 0 | 0 | 0 | 0 |

Table 2: **Win rates (%) across all nine race matchups (Lv.4–Lv.10) in our TEXTSCII-ALL evaluation environment and performance of SoTAs.** Rows sharing the same color shading indicate the same Player vs. Opponent matchup. Whereas prior baselines (Shao et al., 2024; Ma et al., 2024; Wu & Hu, 2025; Li et al., 2025b) each focus on a single matchup, we evaluate HIMA on all player-versus-opponent pairs (Protoss, Terran, Zerg). Each row reports win rates over 50 games per difficulty level; dashes (–) denote untested conditions. Our Terran results are comparatively lower, likely due to its higher micro demands (we primarily target macro control), which we plan to address in future work.

**TEXTSCII-ALL evaluation environment.** We propose a new text-based StarCraft II evaluation environment, called 'TEXTSCII-ALL', which expands upon the existing TextStarCraft II (Ma et al., 2024) environment. While previous work only supported a single matchup—Protoss (player) vs. Zerg (opponent)—our TEXTSCII-ALL evaluation environment includes comprehensive battles across all three races (Protoss, Terran, Zerg) and supports all nine possible player-opponent race combinations. To achieve this, we define and implement complete, race-specific action spaces for Terran and Zerg in addition to Protoss, enabling any race to serve as either the player. This broader action space and enhanced evaluation scope facilitate comprehensive and fair benchmarking. Implementation details and the full definition of each race-specific action space are provided in Appendix Sec. D.

**Evaluation metric.** Ma et al. (2024) propose various evaluation metrics, including win-rate, to assess model performance. However, many of these metrics do not closely correlate with the primary measure of success, namely, the win rate, especially as the game progresses (but we provide further details on these metrics and results in Appendix Sec. F). Consequently, we employ win-rate as our main evaluation criterion under different difficulty levels.

## 4.2 Baselines

We compare our proposed HIMA against recent SoTA approaches: SwarmBrain (Shao et al., 2024), TextStarCraft (Ma et al., 2024), EpicStar (Wu & Hu, 2025), and the very recently proposed HEP (Li et al., 2025b) in both built-in AI and head-to-head matchups.

## 4.3 Match-up Results

**With built-in AI.** We match up HIMA with a built-in AI system in all *nine* race combinations (with Protoss, Terran and Zerg) for the first time in the literature (*cf*. prior arts focus on a single race combination) and summarize the results in Table 2. In all match-ups, our approach exhibits high win rates over multiple difficulty levels. In particular, while HIMA consistently achieves high win rates in Protoss and Zerg matches, its performance is

| Opponent | Ours *vs.* Opponent | Win Rate (%) |
|---|---|---|
| SwarmBrain (Shao et al., 2024) | Protoss *vs.* Zerg | 100 |
| TextStarCraft (Ma et al., 2024) | Protoss *vs.* Protoss | 100 |
| HEP (Li et al., 2025b) | Protoss *vs.* Protoss | 100 |

Table 3: **Match-ups between HIMA (Ours) and SoTAs.** All results use HIMA (Protoss) against the listed opponents, measuring the win rate over 10 consecutive games. Our method achieves a 100% win rate, outperforming all previous approaches.

| Agent Configuration | IL Agent Size | Win Rate (%) at Game Difficulty | | | | | | |
|---|---|---|---|---|---|---|---|---|
| | | Lv.4 | Lv.5 | Lv.6 | Lv.7 | Lv.8 | Lv.9 | Lv.10 |
| Single-Agent (IL Only) (Ma et al., 2024) | 7B | 70 | 25 | 0 | 0 | 0 | 0 | 0 |
| Single-Agent (IL Only) | 7B | 90 | 75 | 25 | 15 | 0 | 0 | 0 |
| Single-Agent (IL + SP) | 7B | 90 | 80 | 35 | 20 | 10 | 0 | 0 |
| Multi-Agent (IL + SP) (**Ours**) | 1.5B × 3 | 100 | 92 | 84 | 82 | 68 | 20 | 16 |

Table 4: **Ablation study of different agent configurations.** We compare four agent configurations across difficulty levels 4–10: *Single-Agent (IL Only)* (Ma et al., 2024)trained on previously provided instruction-tuning data from Ma et al. (2024).; *Single-Agent (IL Only)* trained on our human demonstrations; *Single-Agent (IL + SP)* with Strategic Planner added; and *Multi-Agent (IL + SP)* using three 1.5B agents (total ∼4.5B). Results show our dataset improves low-level performance (first row *vs.* second row), adding SP enhances mid-level success (third row), and the multi-agent framework maintains robustness at higher difficulties (fourth row). Each row in the first three reports shows win rates over 20 games per difficulty level, while the fourth shows rates over 50 games.

lower for Terran. We conjecture that this result stems from the need for more fine-grained microlevel control, which goes beyond the macro-focused scope of our current evaluation environment. Addressing these scenarios remains a key direction for future work.

**With state of the arts.** We match our HIMA with three open source baselines and summarize the results in Table 3. Unfortunately, we cannot compare EpicStar (Wu & Hu, 2025) with ours because it has not yet provided publicly available code. For each opponent, we use the same race setting originally reported to achieve their best performance. We select Protoss for HIMA as our previous experiments with built-in AI showed that HIMA (Protoss) consistently achieved the highest win rate among all opposing races. We measure performance in 10 consecutive games in each match, showing that HIMA achieves a 100% win rate in all cases, showing the effectiveness of our proposed method.

## 4.4 Detailed Analysis

For a detailed analysis, we consider only Protoss (player) *vs.* Zerg (opponent) because it represents one of the most commonly studied matchups in previous work (Ma et al., 2024; Li et al., 2025b; Wu & Hu, 2025) by running 20 randomly sampled matches.

**Agent configurations.** We compare four setups from Lv.4 to Lv.10 and summarize the results in Table 4 as follows. (1) *Single-Agent (IL Only)* (Ma et al., 2024) uses previously available instruction-tuning data from Ma et al. (2024); (2) *Single-Agent (IL Only)* leverages our human demonstration–based dataset, which notably improves performance at lower difficulties; (3) *Single-Agent (IL + SP)* adds a Strategic Planner (SP), which increases mid-level win rates; and *Multi-Agent (IL + SP)* employs three 1.5B agents (total ∼4.5B), comparable to a 7B single agent. Results show that while the single-agent models falter beyond Lv.7, the multi-agent approach remains more resilient, sustaining higher win rates at Lv.8–10.

**Multi-agent clustering criteria.** We cluster instruction-tuning data by (1) *opening strategy*, (2) *advancement tempo*, and (3) *unit composition* to train agents with distinct characteristics.

| Clutering criteria | Win Rate (%) at Game difficulty | | | | | | |
|---|---|---|---|---|---|---|---|
| | Lv.4 | Lv.5 | Lv.6 | Lv.7 | Lv.8 | Lv.9 | Lv.10 |
| Opening strategy | 90 | 60 | 10 | 0 | 0 | 0 | 0 |
| Advancement tempo | 100 | 70 | 55 | 40 | 35 | 0 | 0 |
| Unit composition (**Ours**) | 100 | 92 | 84 | 82 | 68 | 20 | 16 |

Table 5: **Performance comparison of different multi-agent clustering criteria at various game difficulty levels.** We experiment with various clustering criteria and empirically confirm that clustering based on unit composition yields superior performance.

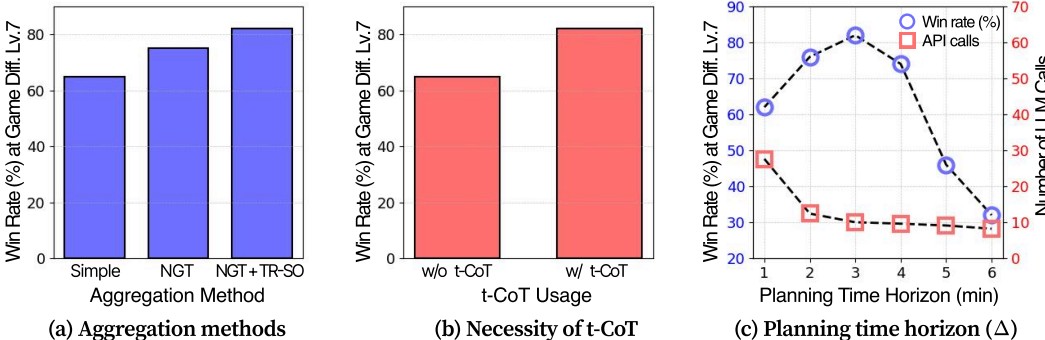

(a) Aggregation methods     (b) Necessity of t-CoT     (c) Planning time horizon (Δ)

Figure 4: **Detailed analyses of the proposed HIMA.** (a) Enhanced aggregation methods improve performance, (b) t-CoT effectively bridges short- and long-term strategies and (c) optimal planning horizon balances reactivity and efficiency.

The opening strategy uses public metadata[1] in early build orders. Advancement tempo differentiates tech-rushing from early-unit production strategies. For the unit composition, we group replays based on the types of unit used in each match-up. As shown in Tab. 5, unit composition clustering performs best, guiding the planner to select optimal units for evolving game conditions (Appendix Sec. E.2 for details). Furthermore, we analyze how response diversity generated through these various clustering approaches impacts overall performance in Appendix Sec. I.

**Aggregation method of the strategic planner.** Figure 4-(a) shows comparative results on three aggregation methods for the SP **Simple** merges action sequences without coordination (*i.e.*, without *advisory strategy resolution* in Sec. 3.2). **NGT** applies Nominal Group Technique (Delbecq & de Ven, 1971) to transform agents' proposals into a unified plan considering the environment. **NGT + TR-SO** extends this by incorporating Tactical Rationale and Strategic Objective from each agent. Results show **NGT + TR-SO** achieves the highest win rate at difficulty level 7, attributable to the synergy of combining tactical justifications with strategic goals, allowing coherent decision-making under dynamic conditions.

**Temporal CoT.** We compare win rates with and without the proposed 'temporal Chain-of-Thought (t-CoT)' and summarize the results in Figure 4-(b). Results show t-CoT outperforms the baseline, indicating that systematically linking immediate, short-term, and long-term goals enhances strategic coherence and responsiveness. The performance gap underscores the importance of temporal reasoning in hierarchical planning for competitive environments.

**Time-window selection for specialized agents.** We illustrates how extending the time window (Δ) for action sequence generation affects both average API calls and win rates (at game difficulty Lv. 7) in Figure 4-(c). With a short planning horizon (*e.g.*, 1 minute), the agent queries the model more often and reacts quickly, but this frequent disruption raises computational overhead without necessarily improving performance. As the horizon

---

[1]https://lotv.spawningtool.com/replays/?pro_only=on

| Method | Time per LLM call | Avg. #LLM calls (in 20m) | Total LLM call time (s) |
|---|---|---|---|
| TextStarCraft (Ma et al., 2024) | 12.1 | 636 | 7,695 |
| HEP (Li et al., 2025b) | 10.4 | 468 | 4,867 |
| HIMA (**Ours**) | 22.5 | 11 | 247 |

Table 6: **LLM call overhead across 20 winning matches.** Each averaging about 20 minutes. While the hierarchical multi-agent framework of HIMA increases the duration of individual LLM calls, its dramatically lower call frequency substantially reduces overall LLM overhead (*e.g.*, 247 s *vs*. several thousand seconds). In real-time play, frequent LLM calls can stall game progression, highlighting the need to minimize total call time.

lengthens, average API calls decline steadily, freeing up resources while still providing enough adaptability for mid-game shifts.

However, beyond a certain midpoint (here, around 3–4 minutes), over-long planning horizons become too rigid, missing critical opportunities to respond to immediate threats. This trade-off produces a peak in the win rate on an intermediate horizon, demonstrating that *moderate* time windows strike the best balance between a timely reaction and a long-term cohesive strategy.

**LLM call overhead.** To compare LLM call overhead, we summarize the average time per LLM call, calls on average 20 minutes, and the total LLM time over 20 matches in Table 6. Despite a longer call time (22.5 seconds) in our proposed HIMA due to its three specialized agents and GPT-4o-mini planner, HIMA makes only 11 calls per 20-minute game through effective long-term planning. This efficiency reduces the total LLM response time to 247s, compared to thousands for other methods. In real-time settings, where game play continues during processing, HIMA's reduced call frequency provides smoother play experience.

## 5 Conclusion

We propose a hierarchical imitation multi-agent (HIMA) framework for SC2, where specialized agents generate structured action sequences and a meta-controller fuses these into cohesive strategies. Using human replay data with tactical rationales, our method preserves build orders, reduces LLM queries, and adapts to dynamic conditions. We also introduce the TEXTSCII-ALL environment, which covers all nine race matches, as a comprehensive RTS testbed. The results show that HIMA achieves better win rates and efficiency compared to SoTA baselines, demonstrating the effectiveness of combining imitation-based specialization with high-level orchestration in complex RTS environments.

## Acknowledgment

This work was partly supported by CARAI grant funded by DAPA and ADD (UD230017TD, 45%) and the IITP grants (No.RS-2022-II220077, No.RS-2022-II220113, No.RS-2022-II220959, No.RS-2022-II220871, No. RS-2025-02263598 (20%), No.RS-2021-II211343 (SNU AI), No.RS-2021-II212068 (AI Innov. Hub), No. RS-2025-25442338 (AI Star Fellowship-SNU)) funded by the Korea government(MSIT).

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

## A  Brief Introduction of The Game of StarCraft II

StarCraft II (SC2), developed by Blizzard Entertainment, is a real-time strategy (RTS) game renowned for its depth, complexity, and strong presence in the e-sports arena. Players choose from one of three race: Terrans (humans), Protoss (technologically advanced aliens), or Zerg (rapidly evolving lifeforms). Each offering unique units and strategies that demand varied approaches to resource management, base building, and tactical combat.

Notably, the level of micro-control (unit-level precision) often ranks Terran at the highest, followed by Zerg, and then Protoss. This trend is also reflected in our benchmark results, where Terran shows the lowest rule-based micro-management performance, Zerg ranks in the middle, and Protoss achieves the highest.

**Key Gameplay Elements**

- **Unit Coordination and Macro-Management:** Selecting appropriate build orders and unit compositions while managing expansions, production cycles, and researching key technologies is crucial for establishing a robust economy and effectively countering the opponent's strategies.
- **Resource Prioritization:** Balancing the collection and allocation of minerals and vespene gas is crucial for effective unit production and technological advancement.
- **Fog of War and Scouting:** Restricted visibility compels players to deploy scouts or specialized units to gather intelligence, adapt strategies, and anticipate opponent actions.
- **Dynamic Battlefield Adaptation:** As the game progresses and the battlefield evolves, players must continuously adjust their tactics to stay ahead of emerging threats and opportunities.

## B  Previous Work: TextStarCraft II (Ma et al., 2024)

TextStarCraft II (Ma et al., 2024) is the first environment for evaluating LLM-based agents in real-time strategic scenarios within StarCraft II. By leveraging a Chain of Summarization (CoS) technique to enhance rapid decision-making, the authors demonstrate that most tested LLMs surpass Level 5 built-in AI.

## C  Implementation Details and Experimental Hardware Setup

### C.1  Environment and Game Settings

**Game Version** : We conduct all experiments using Patch 5.0.14.93333 of StarCraft II. We use the built-in AI provided in the game, and the difficulty settings are detailed in Table 7.

| Game Difficulty Level | Blizzard Difficulty |
|:---:|:---:|
| 1 | Very Easy |
| 2 | Easy |
| 3 | Medium |
| 4 | Hard |
| 5 | Harder |
| 6 | Very Hard |
| 7 | Elite |
| 8 | Cheat Vision |
| 9 | Cheat Money |
| 10 | Cheat Insane |

Table 7: **StarCraft II Built-in AI Difficulty Levels.**

**Map Selection**: All evaluations are performed on the Ancient Cistern LE maps from the 2023 StarCraft II esports 1v1 ladder.

**Model Temperature**: To allow for a moderate level of diversity in responses, we set the temperature parameter to 0.7 for both the imitation agents and the strategic planner.

## C.2 Experimental Hardwares

**Training**: The imitation agents are fine-tuned on four NVIDIA H100 80GB GPUs, requiring approximately six hours in total. During fine-tuning, LoRA modules are added for efficient adaptation of each agent.

**Inference**: We use a single NVIDIA A6000 40GB GPU to concurrently run three 1.5B-parameter imitation agents during evaluation.

# D    Details About TEXTSCII-ALL Evaluation Environment

The code is based on the python-sc2[2] library. Python-sc2 is an open-source Python framework that allows developers to create StarCraft II AI bots by providing convenient APIs for controlling and observing the game environment.

## D.1    Agent vs. Built-in AI Matches

We categorize the action space into three groups—unit production, building construction, and technology development (see Fig. 6 for details). The Protoss have 58 possible actions, the Zerg 61, and the Terrans 62. Race-specific commands (*e.g.*, Chrono Boost, Larva Inject, Mule) and a few general commands (such as "attack" or "scout") operate according to rule-based logic. We implement each of these actions within our codebase, performing prerequisite checks (resource availability, required structures, etc.) before execution.

## D.2    Agent vs. Agent Matches (Agent Arena)

In addition to built-in AI matches, we conduct head-to-head encounters ("Agent Arena") under the same map settings and resource conditions. Because both agents operate within the same constraints and action space, these matches enable a direct comparison of strategic reasoning.

# E    Details About Data Construction and Clustering for Imitation Learning

## E.1    Details About Instruction-Tuning Dataset Construction

From SC2EGSET(Białecki et al., 2023), we identify each time step $t$ at which an action $A_t$ (belonging to our defined action space) is issued. We then record the player's current state $S_t$ at time $t$, which includes supply details, unit details, building details, technology details, and ongoing commands. Notably, $S_t$ is limited to the player's information only and does not contain any enemy-related data.

At each decision point $t$, we pair the state $S_t$ with the action $A_t$ executed at time $t$, as well as all subsequent actions occurring within the next $\Delta$ minutes, forming the action sequence $A_{t:t+\Delta}$. This $(S_t, A_{t:t+\Delta})$ pair constitutes one training sample. Empirically, setting $\Delta = 3$ yields the best performance (see Figure 4-(c)). Once this action sequence is constructed, we shift the window to the time of the next executed action $t'$ (i.e., the first action after $t$), and again collect all actions that occur between $t'$ and $t' + \Delta$. By repeating this sliding-window procedure, we obtain on average about 200 training samples per game. A detailed overview of this data construction process is provided in Figure 5.

---

[2]https://github.com/BurnySc2/python-sc2

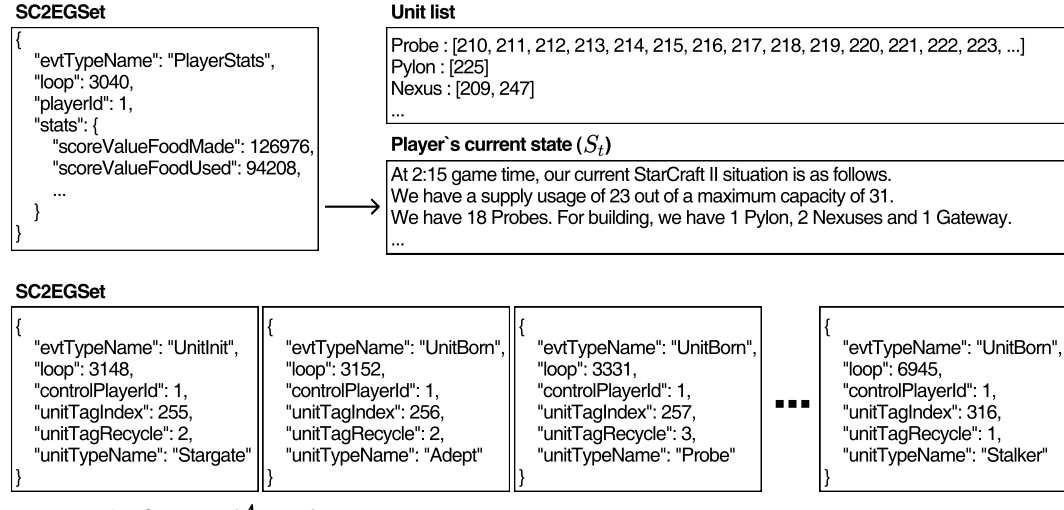

Figure 5: **Overview of data construction process.** We use SC2EGSET to obtain player's current state and corresponding action sequence. In particular, the number of units in each current state is determined by separately recording unit counts at each time step.

In order to convert the action-only dataset produced by the above procedure into an instruction-tuning format that includes rationales, we use a specialized system prompt alongside the GPT-4o-mini API to generate the reasons behind each decision point's action sequence. The system prompt guides the model to break the rationale into three time frames—immediate, short-term, and long-term strategy (see Figure 7). By training on this enriched dataset, the model adopts these three perspectives to more precisely interpret the game environment, resulting in more strategic decisions.

## E.2 Details About Imitation Multi-Agent Clustering

We explore multiple criteria for creating specialized agents. Among these, we generate an imitation multi-agent system based on army ratios, thereby broadening the range of possible playstyles and strategies. However, we also experimented with alternative criteria, which did not yield as clear a differentiation. We include them here for completeness.

**Army Composition Criterion.** To encourage each imitation agent to develop a unique style, we apply $k$-means clustering based on the proportion of units used by each race. By focusing on each agent's unit ratio, we reveal clear distinctions among them. For instance, when clustering Protoss agents, they naturally split into three groups: one that relies heavily on Phoenixes and Colossi, another centered around Zealots and High Templars, and a third emphasizing Carriers and Void Rays.

**Unit Ratio Extraction and Clustering.** To capture meaningful differences in army compositions, we focus on games that the professional player won and calculate a weighted unit distribution. Specifically, for each game, we multiply the count of each unit type by its corresponding "supply cost" (or "population value") to account for the greater impact of high-tier units. We then sum these supply-adjusted counts and normalize each unit type by this total to obtain a distribution vector of length $|\mathcal{U}|$, where $\mathcal{U}$ denotes the set of unit types for a given race. Formally, if $n_i$ is the count of unit type $i$ and $w_i$ is the supply cost, then the normalized ratio $r_i$ is given by:

$$r_i = \frac{n_i \times w_i}{\sum_{j \in \mathcal{U}} \left( n_j \times w_j \right)}.$$

```
Protoss Action Space

'TRAIN UNIT': {
    0: 'PROBE', 1: 'ZEALOT', 2: 'ADEPT', 3: 'STALKER', 4: 'SENTRY', 5: 'HIGHTEMPLAR', 6: 'DARKTEMPLAR', 7: 'VOIDRAY',
    8: 'CARRIER', 9: 'TEMPEST', 10: 'ORACLE', 11: 'PHOENIX', 12: 'MOTHERSHIP', 13: 'OBSERVER', 14: 'IMMORTAL',
    15: 'WARPPRISM', 16: 'COLOSSUS', 17: 'DISRUPTOR', 18: 'ARCHON'
},
'BUILD STRUCTURE': {
    19: 'PYLON', 20: 'ASSIMILATOR', 21: 'NEXUS', 22: 'GATEWAY', 23: 'CYBERNETICSCORE', 24: 'FORGE', 25: 'TWILIGHTCOUNCIL',
    26: 'ROBOTICSFACILITY', 27: 'STARGATE', 28: 'TEMPLARARCHIVE', 29: 'DARKSHRINE', 30: 'ROBOTICSBAY',
    31: 'FLEETBEACON', 32: 'PHOTONCANNON', 33: 'SHIELDBATTERY'
},
'RESEARCH TECHNIQUE': {
    34: 'WARPGATE_RESEARCH', 35: 'AIRWEAPONS_LEVEL1', 36: 'AIRWEAPONS_LEVEL2', 37: 'AIRWEAPONS_LEVEL3',
    38: 'AIRARMORS_LEVEL1', 39: 'AIRARMORS_LEVEL2', 40: 'AIRARMORS_LEVEL3', 41: 'ADEPT_RESONATING_GLAIVES',
    42: 'STALKER_BLINK', 43: 'ZEALOT_CHARGE', 44: 'GROUNDWEAPONS_LEVEL1', 45: 'GROUNDWEAPONS_LEVEL2',
    46: 'GROUNDWEAPONS_LEVEL3', 47: 'GROUNDARMORS_LEVEL1', 48: 'GROUNDARMORS_LEVEL2',
    49: 'GROUNDARMORS_LEVEL3', 50: 'SHIELDS_LEVEL1', 51: 'SHIELDS_LEVEL2', 52: 'SHIELDS_LEVEL3',
    53: 'COLOSSUS_EXTENDED_THERMAL_LANCE', 54: 'WARPPRISM_GRAVITIC_DRIVE', 55: 'OBSERVER_GRAVITIC_BOOSTERS',
    56: 'HIGHTEMPLAR_PSISTORM', 57: 'VOIDRAY_SPEED_UPGRADE', 58: 'PHOENIX_RANGE_UPGRADE'
}
```

```
Zerg Action Space

'TRAIN UNIT': {
    0: 'DRONE', 1: 'OVERLORD', 2: 'QUEEN', 3: 'ZERGLING', 4: 'ROACH', 5: 'HYDRALISK', 6: 'MUTALISK', 7: 'CORRUPTOR',
    8: 'INFESTOR', 9: 'SWARMHOST', 10: 'ULTRALISK', 11: 'VIPER', 12: 'BANELING', 13: 'RAVAGER', 14: 'LURKER',
    15: 'BROODLORD', 16: 'OVERSEER'
},
'BUILD STRUCTURE': {
    17: 'EXTRACTOR', 18: 'HATCHERY', 19: 'SPAWNINGPOOL', 20: 'BANELINGNEST', 21: 'ROACHWARREN',
    22: 'HYDRALISKDEN', 23: 'LAIR', 24: 'HIVE', 25: 'EVOLUTIONCHAMBER', 26: 'INFESTATIONPIT', 27: 'SPIRE',
    28: 'GREATERSPIRE', 29: 'ULTRALISKCAVERN', 30: 'LURKERDEN', 31: 'SPORECRAWLER', 32: 'SPINECRAWLER'
},
'RESEARCH TECHNIQUE': {
    33: 'ZERGLING_SPEED_UPGRADE', 34: 'ZERGLING_ATTACKSPEED_UPGRADE', 35: 'BANELING_SPEED_UPGRADE',
    36: 'ROACH_SPEED_UPGRADE', 37: 'ROACH_TUNNELING_CLAWS', 38: 'OVERLORD_SPEED_UPGRADE', 39: 'BURROW',
    40: 'HYDRALISK_SPEED_UPGRADE', 41: 'HYDRALISK_RANGE_UPGRADE', 42: 'MELEEWEAPONS_LEVEL1',
    43: 'MELEEWEAPONS_LEVEL2', 44: 'MELEEWEAPONS_LEVEL3', 45: 'MISSILEWEAPONS_LEVEL1',
    46: 'MISSILEWEAPONS_LEVEL2', 47: 'MISSILEWEAPONS_LEVEL3', 48: 'GROUNDARMORS_LEVEL1',
    49: 'GROUNDARMORS_LEVEL2', 50: 'GROUNDARMORS_LEVEL3', 51: 'INFESTOR_NEURAL_PARASITE',
    52: 'FLYERWEAPONS_LEVEL1', 53: 'FLYERWEAPONS_LEVEL2', 54: 'FLYERWEAPONS_LEVEL3', 55: 'FLYERARMORS_LEVEL1',
    56: 'FLYERARMORS_LEVEL2', 57: 'FLYERARMORS_LEVEL3', 58: 'ULTRALISK_ARMOR_UPGRADE',
    59: 'ULTRALISK_SPEED_UPGRADE', 60: 'LURKER_RANGE_UPGRADE', 61: 'LURKER_DIGGING_CLAWS'
}
```

```
Terran Action Space

'TRAIN UNIT': {
    0: 'SCV', 1: 'MARINE', 2: 'REAPER', 3: 'MARAUDER', 4: 'GHOST', 5: 'HELLION', 6: 'WIDOWMINE', 7: 'CYCLONE', 8: 'SIEGETANK',
    9: 'THOR', 10: 'VIKING', 11: 'MEDIVAC', 12: 'LIBERATOR', 13: 'BANSHEE', 14: 'RAVEN', 15: 'BATTLECRUISER'
},
'BUILD STRUCTURE': {
    16: 'COMMANDCENTER', 17: 'SUPPLYDEPOT', 18: 'REFINERY', 19: 'BARRACKS', 20: 'ENGINEERINGBAY', 21: 'BUNKER',
    22: 'FACTORY', 23: 'STARPORT', 24: 'ORBITALCOMMAND', 25: 'PLANETARYFORTRESS', 26: 'BARRACKSREACTOR',
    27: 'BARRACKSTECHLAB', 28: 'FACTORYREACTOR', 29: 'FACTORYTECHLAB', 30: 'STARPORTREACTOR',
    31: 'STARPORTTECHLAB', 32: 'GHOSTACADEMY', 33: 'ARMORY', 34: 'FUSIONCORE', 35: 'MISSILETURRET',
    36: 'SENSORTOWER'
},
'RESEARCH TECHNIQUE': {
    37: 'STIMPACK', 38: 'MARINE_HEALTH_UPGRADE', 39: 'HELLION_ATTACK_UPGRADE', 40: 'CYCLONE_ATTACK_UPGRADE',
    41: 'WIDOWMINE_DRILLING_CLAWS', 42: 'SMARTSERVOS', 43: 'BANSHEE_CLOAKING_FIELD',
    44: 'BANSHEE_SPEED_UPGRADE', 45: 'MEDIVAC_RAPID_REIGNITION_SYSTEM', 46: 'INFANTRYWEAPONS_LEVEL1',
    47: 'INFANTRYWEAPONS_LEVEL2', 48: 'INFANTRYWEAPONS_LEVEL3', 49: 'INFANTRYARMORS_LEVEL1',
    50: 'INFANTRYARMORS_LEVEL2', 51: 'INFANTRYARMORS_LEVEL3', 52: 'NEOSTEEL_ARMOR', 53: 'HI-SEC_AUTO_TRACKING',
    54: 'GHOST_CLOAKING', 55: 'VEHICLEWEAPONS_LEVEL1', 56: 'VEHICLEWEAPONS_LEVEL2', 57: 'VEHICLEWEAPONS_LEVEL3',
    58: 'SHIPWEAPONS_LEVEL1', 59: 'SHIPWEAPONS_LEVEL2', 60: 'SHIPWEAPONS_LEVEL3',
    61: 'BATTLECRUISER_REFIT', 62: 'LIBERATOR_RANGE_UPGRADE'
}
```

Figure 6: **Action space of three races in StarCraft II.** Each race has actions related to unit production, building construction, and technology development. Both the imitation agent and the strategic planner form sequences of these race-specific actions to issue commands.

---

**SYSTEM**
You are a bot with extensive StarCraft II Protoss expertise. You are given the current game state and the professional player's chosen actions for the next 3 minutes. Based solely on these details, provide concise coaching advice in three lines - one line each for immediate steps, short-term actions, and long-term strategy - in order of priority. Include specific units and their numbers where possible, and do not mention or comment on any actions currently waiting.

**USER**
At 04:48 game time, our current StarCraft II situation is as follows. We have a supply usage of 39 out of a maximum capacity of 62. We have 30 Probes, 1 Zealot and 1 Phoenix. For buildings, we have 2 Nexuses, 5 Pylons, 4 Assimilators, 1 Gateway, 1 Cyberneticscore, 1 Forge, 1 Stargate and 4 Photoncannons. No research has been completed yet. We are waiting 1 Probe, 1 Oracle and 2 Stargates. <TRAIN PROBE> x 3 <TRAIN ORACLE> <TRAIN PROBE> x 3 <TRAIN ORACLE> <TRAIN PROBE> <BUILD PYLON> <TRAIN PROBE> <TRAIN ORACLE> <TRAIN PROBE> x 2 <TRAIN ORACLE> <TRAIN PROBE> <BUILD PYLON> <TRAIN PROBE> x 2 <TRAIN ORACLE> <BUILD PYLON> <BUILD NEXUS> <TRAIN ORACLE> x 2 <BUILD PYLON> x 2 <TRAIN PROBE> x 2 <TRAIN ORACLE> <BUILD PYLON> <BUILD PHOTONCANNON> <BUILD PYLON> <TRAIN ORACLE> <TRAIN PROBE> x 2 <TRAIN ORACLE> <TRAIN PROBE> x 2 <TRAIN ORACLE> x 2 <TRAIN PROBE> x 3 <BUILD ASSIMILATOR> x 2

**ASSISTANT**
**Immediate Steps**: Focus on maximizing your economy by training Probes consistently; aim for 40+ Probes to ensure a strong resource income.
**Short-term Actions**: Prioritize building additional Pylons to avoid supply block and ensure continuous production of units, particularly Oracles for harassment and scouting.
**Long-term Strategy**: Transition into a robust army composition by teching towards Robotics Facility or additional Gateways for ground units, while maintaining Oracle production for map control and harassment.

Figure 7: **Prompt Details for Rational Generation in an Instruction-Tuning Dataset.** We use GPT-4o-mini to generate the rationale behind the chosen action sequence for each situation. Subsequently, the generated rationale and the corresponding action sequence are combined into a single output within our dataset.

Finally, we perform $k$-means clustering on these vectors in the unit-ratio space, grouping together games with similar army compositions. This process yields distinct army-style clusters and allows each specialized agent to learn a different strategic focus.

**Opening Strategy Criterion.** We obtain early-game build orders from publicly available metadata[3]. Among the available tags, we focus on those containing the word "opening" (e.g., for Protoss, "Oracle opening," "Phoenix opening," etc.) to categorize the corresponding games, under the assumption that such tags effectively capture each game's initial strategy. However, because these tags are arbitrarily assigned by users rather than following consistent criteria, this approach underperforms compared to clustering by army composition, and no clearly distinguishable characteristics emerge for each imitation agent.

**Advancement Tempo Criterion.** By examining when key buildings and technologies are constructed for each race (e.g., for Protoss, "WarpgateResearch," "Fleetbeacon," etc.), we use $k$-means clustering to classify each game as slow, moderate, or fast in terms of progression speed. Because advancement tempo is heavily influenced by resource fluctuations and in-game events, it reflects only the pace of development rather than a player's overall strategy. By contrast, final unit ratios capture the compositions that players ultimately commit to, thereby revealing clearer strategic distinctions.

### E.3    Details About Dataset Clustering

After clustering the dataset according to the Army Composition Criterion, each imitation agent is assigned a subset of the data. Specifically, for each race we obtain the following distribution (in thousands of samples) across the three agents:

- **Protoss**: 11.5k (Agent A), 13.2k (Agent B), 15.2k (Agent C)
- **Zerg**: 12.1k (Agent A), 17.9k (Agent B), 9.8k (Agent C)
- **Terran**: 9.5k (Agent A), 11.0k (Agent B), 10.4k (Agent C)

---

[3] https://lotv.spawningtool.com/replays/?pro_only=on

## F  Details about Evaluation Metrics (Ma et al., 2024)

We also evaluate the performance using the APU, RUR, PBR, and TR metrics proposed in Ma et al. (2024), as shown in Tab. 8.

- **PBR (Production-Block Ratio)**: Calculates the ratio of time spent at maximum supply (200/200) to the total time, where a higher value indicates less efficient macro-management.

- **RUR (Resource Utilization Ratio)**: Measures the total resource expenditure (mineral, gas) until the agent first reaches maximum supply, with a higher value implying weaker macro-strategic use of resources.

- **APU (Average Population Utilization)**: Averages the ratio of used population to the supply cap until maximum supply, where a higher value reflects more efficient macromanagement.

- **TR (Technology Rate)**: Calculates the proportion of completed technologies to the total available, where a higher value suggests a greater emphasis on technological advancement.

Compared to previous studies, the RUR value is relatively high, reflecting the frequent use of more expensive units, whereas the TR value is lower because only essential technologies are selectively upgraded. These slightly higher RUR and lower TR values result from an intentional strategy that prioritizes powerful units and focuses upgrades on what is truly necessary for effective combat. Notably, our method still achieves superior APU and PBR values, indicating efficient resource utilization and effective macromanagement. This combination ultimately leads to a higher win rate, demonstrating that strong unit composition and targeted upgrades enhance overall performance.

However, these metrics have limitations. PBR can become misleadingly high if battles continue after the army has reached maximum supply, which doesn't necessarily mean the player is managing resources poorly. RUR penalizes compositions with expensive units, even though such choices can drive higher win rates. Finally, TR can be lower when only the most essential technologies are researched, yet upgrading everything without a clear plan can also undermine a winning strategy. Hence, relying solely on these metrics may not fully capture the breadth of LLM agents' strategic decision-making.

| Method | Win rate (%) on Lv.5. | PBR ($\downarrow$) | RUR ($\downarrow$) | APU ($\uparrow$) | TR ($\uparrow$) |
|---|---|---|---|---|---|
| TextStarCraft (Ma et al., 2024) | 55 | 0.0781 | 7875 | 0.7608 | 0.4476 |
| EpicStar (Wu & Hu, 2025) | 67 | 0.1211 | 9864 | 0.8123 | 0.2536 |
| **HIMA (Ours)** | 92 | 0.0547 | 15525 | 0.8325 | 0.2233 |

Table 8: **Performance on Additional Evaluation Metrics at Harder Level (Lv.5).** We present results for Win rates (%), APU, RUR, PBR, and TR against Computers on Harder Level (Lv.5). Note that lower values indicate better performance for PBR and RUR, while higher values are better for APU and TR. The reported metrics for *TextStarCraft* and *EpicStar* are taken from the *EpicStar* paper.

## G  Details about the Number of Specialized Agents

Table 9 shows the results of evaluating HIMA at various game difficulties while varying the number of specialized agents within the architecture. As the number of agents increases, overall performance initially improves but eventually saturates, and beyond a certain point, it starts to decline. We conjecture that once a critical threshold is reached, the added complexity and coordination overhead among too many agents can negatively impact overall efficiency and decision making.

| Number of agents | Win Rate (%) at Game Difficulty | | | | | | |
|:---:|:---:|:---:|:---:|:---:|:---:|:---:|:---:|
| | Lv.4 | Lv.5 | Lv.6 | Lv.7 | Lv.8 | Lv.9 | Lv.10 |
| 1 | 90 | 80 | 35 | 20 | 10 | 0 | 0 |
| 2 | 100 | 80 | 55 | 20 | 10 | 10 | 10 |
| 3 | 100 | 92 | 84 | 82 | 68 | 20 | 16 |
| 4 | 100 | 85 | 80 | 70 | 55 | 10 | 10 |
| 5 | 100 | 85 | 80 | 75 | 60 | 15 | 10 |

Table 9: **Win rate by number of agents.** We report the Win rates (%) of multi-agent systems with varying numbers of agents across different StarCraft II built-in AI difficulty levels.

## H    Number of Executed Actions over Gameplay in Human Demonstration

In the early stages of the game, players face constraints imposed by limited resources and the requirement to unlock specific technologies. However, as the match progresses and economies and technology trees expand, players gain access to a much wider variety of tactics. This progression is evident in professional gameplay data (Fig. 8), which shows a steady rise in the number of executed actions up to around the 12-minute mark.

Unlike existing methods (Ma et al., 2024; Wu & Hu, 2025; Li et al., 2025b) that often produce frequent, isolated actions at each timestep (leading to infeasible or redundant commands), HIMA generates structured, multi-step sequences that preserve build orders and maintain strategic coherence over longer horizons. By training on professional demonstrations, HIMA captures the evolving pattern of available actions and, through a flexible rather than fixed-length mechanism, incorporates long-term actions from the current state. This flexibility more accurately reflects the dynamic spikes in action frequency throughout a match, ensuring that both short-term tactics and extended strategic developments are properly modeled.

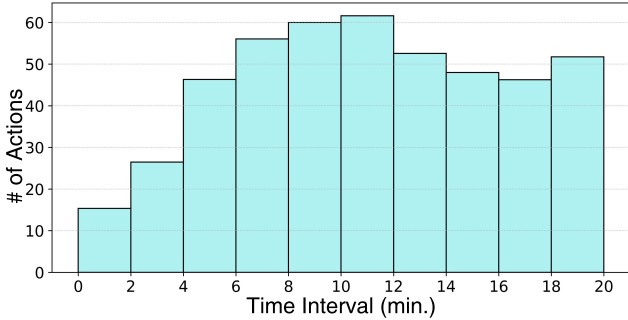

Figure 8: **Number of actions generated per time interval.** We compare the number of actions generated per time interval across 68 games from the Star League 7 tournament held in 2021. We observe that early stages produce fewer actions due to limited resources and tech prerequisites, while later stages result in a higher number of actions enabled by a broader set of tactical options.

## I    In-depth Analysis of Multi-Agent Response Diversity

We measure the diversity of multi-agent outputs using a negative log-likelihood metric from prior work (Subramaniam et al., 2025), where a higher value indicates greater variability among the generated actions. As shown in Figure 9, methods that produce more diverse

responses also tend to achieve higher overall performance. This observation aligns with the "society of mind" principle, which suggests that incorporating multiple viewpoints can foster balanced decision-making in dynamic, real-time strategy settings. In particular, the "unit composition" approach—explicitly designed to promote diverse agent outputs—yields the greatest variety and achieves the highest performance. These findings indicate that encouraging a wider range of agent actions can lead to more robust and effective strategic outcomes.

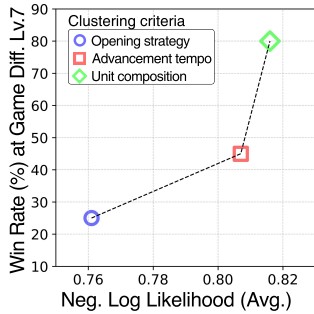

Figure 9: **Agent response diversity across different clustering criteria.** We measure the diversity of multi-agent outputs across opening strategy, advancement tempo, and unit composition criteria.

## J  Performance Impact of Rationales in the Instruction-Tuning Dataset

In Table 10, we compare the performance of IL-only single-agent systems with and without rationales, where these rationales can be viewed as a form of chain-of-thought (Wei et al., 2022). By explicitly providing intermediate reasoning steps, the model can break down complex decisions into smaller sub-decisions before reaching a final action. This process enhances the imitation agent's internal reasoning and yields more coherent multi-step planning, which is crucial for strategic domains like real-time strategy games. Specifically, the model learns to balance immediate, short-term, and long-term strategic needs, thereby enabling more robust strategies across different phases of the game.

The data in Table 10 demonstrate that including rationales leads to higher win rates across various difficulty levels (Lv.4–Lv.10). This highlights the value of interpretable rationales in guiding the agent's decisions.

|  | **Win Rate (%) at Game Difficulty** | | | | | | |
|---|---|---|---|---|---|---|---|
|  | **Lv.4** | **Lv.5** | **Lv.6** | **Lv.7** | **Lv.8** | **Lv.9** | **Lv.10** |
| Decision | 80 | 60 | 10 | 10 | 0 | 0 | 0 |
| Rational + Decision | 90 | 75 | 25 | 15 | 0 | 0 | 0 |

Table 10: **Win rates depending on the presence of rationales.** We report the win rates (%) of IL-only single-agent systems with and without rationales on our instuction-tuning dataset.

## K  Details of Generating Imitation Agent's Strategic Objective

After clustering the data based on unit ratios, use the clear differences among those clusters to define a strategic objective for each one according to its unique unit ratio pattern. We use the chatgpt-o1 model to examine each cluster's unit ratio pattern, guiding us to infer the corresponding strategic objectives. Regardless of race, when dividing the data into three clusters, each cluster is categorized respectively as a ground + support unit focus, an air unit

focus, or a hybrid of ground and air units. The prompt for generating strategic objectives and the strategic objectives for each race are shown in the Figure 10.

**System Prompt**

Here are the ratio distributions for 15 Protoss units in a specific composition for Advisor A, B, and C.

['ZEALOT', 'ADEPT', 'STALKER', 'SENTRY', 'HIGHTEMPLAR', 'DARKTEMPLAR', 'VOIDRAY',  'CARRIER', 'TEMPEST', 'ORACLE', 'PHOENIX', 'IMMORTAL', 'COLOSSUS', 'DISRUPTOR', 'ARCHON']
Advisor A:
[0.06, 0.15, 0.01, 0.01, 0.00, 0.00, 0.02, 0.27, 0.05, 0.01, 0.27, 0.02, 0.09, 0.04, 0.00]
Advisor B:
[0.23, 0.09, 0.11, 0.03, 0.10, 0.00, 0.01, 0.02, 0.00, 0.20, 0.03, 0.10, 0.00, 0.02, 0.05]
Advisor C:
[0.09, 0.09, 0.01, 0.01, 0.00, 0.00, 0.40, 0.31, 0.02, 0.03, 0.01, 0.00, 0.00, 0.03, 0.00]

Based on these ratios, please describe each advisor's strategic objectives or overall game plan in a concise bullet-point format, with exactly three bullet points per advisor. Each bullet point should be on a single line and offer a brief rationale for why these units are chosen.
Limit each advisor's explanation to exactly three lines (one line per bullet).
Focus on explaining how the composition emphasizes certain unit types, how they synergize, and what stage of the game they excel in.

**Protoss`s Strategic Objective**

Advisor A:
- A mix of air and ground units, with a clear focus on air dominance (Carriers, Phoenix, Tempests).
- Minimal investment in support or tech units like Sentries, Archons.
- Reflects a late-game hybrid approach, blending air power with limited ground and utility units.

Advisor B:
- Ground-heavy composition with strong frontline units like Zealots, Stalkers, and High Templars.
- Includes versatile harassment and support units like Oracles, Warp Prisms, and Archons for flexibility.
- A balanced and adaptable mid-game strategy that allows for transitions based on the situation.

Advisor C:
- Overwhelming focus on air units, particularly Voidrays and Carriers, with minimal ground presence.
- Almost no investment in support units or utility, emphasizing raw offensive power.
- Represents a late-game air superiority strategy, designed to dominate the skies with high DPS.

**Zerg`s Strategic Objective**

Advisor A:
- A mix of air and ground units, with a clear focus on air dominance (Mutalisks, Corruptors).
- Minimal investment in support or tech units like Infestors and Vipers.
- Reflects a late-game hybrid approach, blending air power with limited ground and utility units.

Advisor B:
- Ground-heavy composition with strong frontline units like Zerglings, Roaches, and Hydralisks.
- Includes versatile harassment and support units like Banelings, Mutalisks, Nydus Worms, or Infestors.
- A balanced and adaptable mid-game strategy that allows transitions into advanced air units or Ultralisks.

Advisor C:
- Overwhelming focus on air units, particularly Brood Lords and Corruptors, with minimal ground presence.
- Almost no investment in support units or utility, emphasizing raw offensive power.
- Represents a late-game air superiority strategy, designed to dominate the skies with high DPS.

**Terran`s Strategic Objective**

Advisor A:
- A mix of air and ground units, with a clear focus on air dominance (Battlecruisers, Liberators, Vikings).
- Minimal investment in support or tech units like Ravens and Ghosts.
- Reflects a late-game hybrid approach, blending air power with limited ground and utility units.

Advisor B:
- Ground-heavy composition with strong frontline units like Marines, Marauders, and Siege Tanks.
- Includes versatile harassment and support units like Medivacs, Reapers, or Hellions.
- A balanced and adaptable mid-game strategy that allows for transitions based on the situation.

Advisor C:
- Overwhelming focus on air units, particularly Battlecruisers and Liberators, with minimal ground presence.
- Almost no investment in support units or utility, emphasizing raw offensive power.
- Represents a late-game air superiority strategy, designed to dominate the skies with high DPS.

Figure 10: **Prompt for generating strategic objectives and strategic objective prompt for each race.** We identify the underlying strategic objective of each imitation agent based on its unit ratio.

## L   Various Open-source Models for Imitation Learning Agent

We conduct additional experiments with various small agent models beyond Qwen-2 1.5B, and the results are summarized in Table 11.  These results show that all tested models achieve similar performance in our setting, even though their performance differences may be more noticeable on other benchmarks (Yang et al., 2024; 2025; Touvron et al., 2023). We believe this is because all models are trained on the same imitation learning data we collected, which minimizes performance gaps between them.

| Model | IL Agent Size | Win-rate (%) at Lv.7 |
|---|---|---|
| Qwen-2 | 1.5B $\times$ 3 | 82 (41/50) |
| Qwen-3 | 1.7B $\times$ 3 | 85 (17/20) |
| Llama-3.2 | 3B $\times$ 3 | 85 (17/20) |

Table 11: Win-rate comparison at difficulty Lv.7 for different imitation agent models.

## M Various Open-source and Closed-source Models for Strategic Planner

To investigate the generalizability of SP across various models, we evaluate multiple open-source and closed-source models, as shown in Table 12. The closed-source GPT-4o model achieves the highest performance. Among the open-source models, larger variants (*e.g.*, Qwen-2.5 72B) perform well, closely approaching the performance of certain closed-source models (*e.g.*, GPT-4o-mini and Claude).

| Model | Size | Type | Win-rate (%) at Lv.7 |
|---|---|---|---|
| Qwen-2.5 | 32B | | 60 (12/20) |
| Qwen-2.5 | 72B | | 75 (15/20) |
| Qwen-3 | 8B | Open-source | 15 (3/20) |
| Qwen-3 | 32B | | 55 (11/20) |
| Llama-3.3 | 70B | | 65 (13/20) |
| GPT-4o-mini | - | | 82 (41/50) |
| GPT-4o | - | Closed-source | 90 (18/20) |
| Claude-sonnet | - | | 70 (14/20) |
| Claude-haiku | - | | 60 (12/20) |

Table 12: Win-rate comparison at difficulty Lv.7 across various models.

# N   Prompt Details

We present the prompts (shown in Fig. 11, 12, and 13) that illustrate how our HIMA agent can generate reasonable strategies and decisions within the StarCraft II environment.

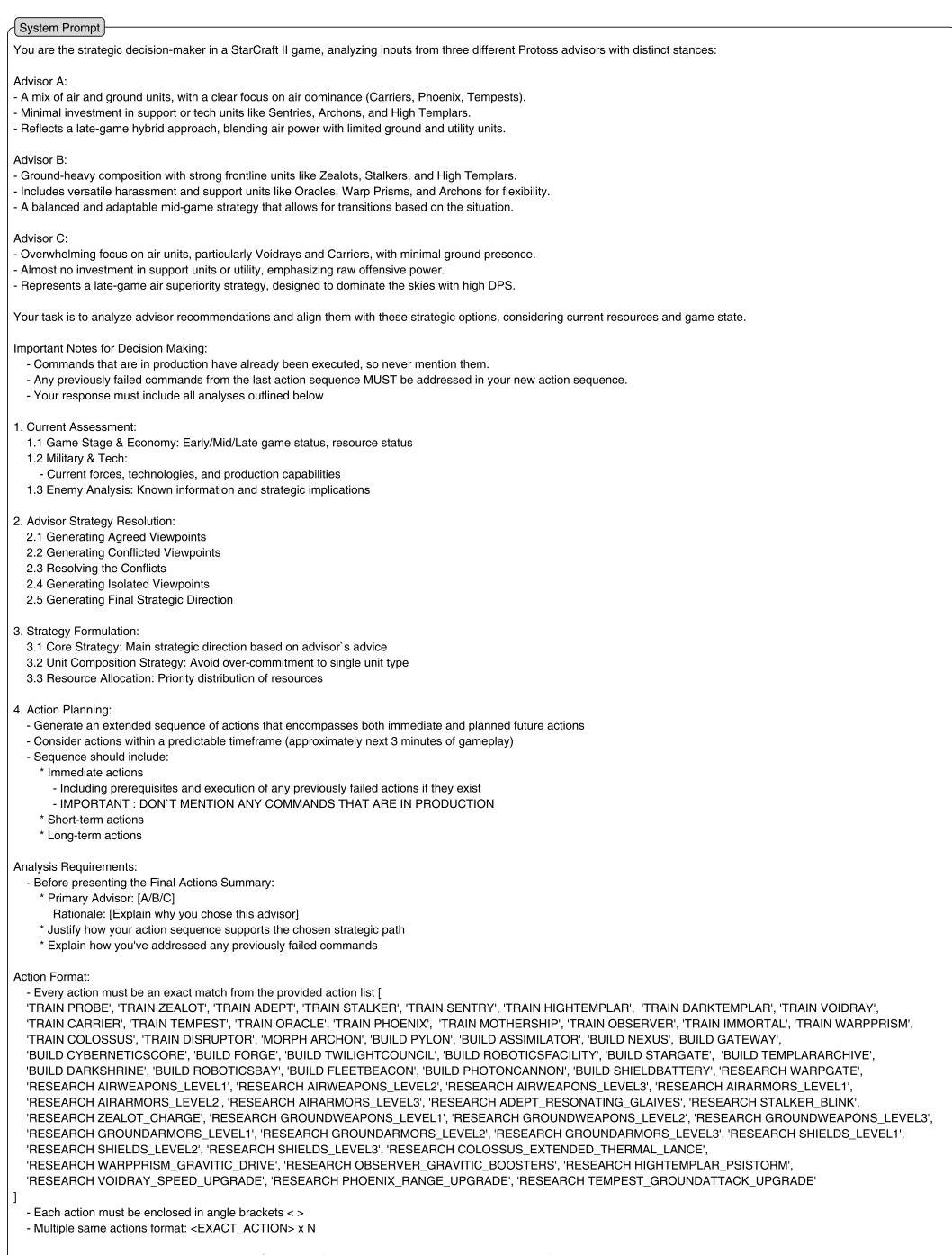

Figure 11: **System prompt for the Protoss race.** It consists of two parts: one that guides each imitation agent's strategic objective, and another that supports the reasoning process of the strategic planner. For other races, the system prompt remains unchanged, except for the list of available actions.

---

**Imitation Agent`s Input**

At 6:32 game time, our current StarCraft II situation is as follows. We have a supply usage of 87 out of a maximum capacity of 109.
We have 46 Probes, 2 Zealots, 5 Adepts, 1 Stalker and 1 Voidray.
For buildings, we have 3 Nexuses, 8 Pylons, 3 Assimilators, 2 Gateways, 1 Stargate and 1 Cyberneticscore.
We have researched Warpgate. We are waiting 2 Probes, 1 Zealot, 1 Adept, 1 Voidray.

---

**Strategic Planner`s Input**

At 6:32 game time, our current StarCraft II situation is as follows. We have a supply usage of 87 out of a maximum capacity of 109.
We have 46 Probes, 2 Zealots, 5 Adepts, 1 Stalker and 1 Voidray.
For buildings, we have 3 Nexuses, 8 Pylons, 3 Assimilators, 2 Gateways, 1 Stargate and 1 Cyberneticscore.
We have researched Warpgate. We are waiting 2 Probes, 1 Zealot, 1 Adept, 1 Voidray.

The advisors provided different suggestions:
advisor A suggested
**Immediate Steps**: Focus on stabilizing your economy by training more Probes (up to 60) to ensure a strong resource income for future unit production.
**Short-Term Actions**: Transition into a carrier-heavy composition by continuously producing Voidrays and Carriers while also building additional Pylons to avoid supply block.
**Long-Term Strategy**: Aim for a robust air force with 5-7 Carriers supported by Adepts and Sentries to counter ground threats while expanding your economy with more Nexuses and Assimilators.

So my advice is <TRAIN PROBE> x 4 <BUILD PYLON> <TRAIN PROBE> <TRAIN VOIDRAY> <BUILD ASSIMILATOR> <TRAIN PROBE> <TRAIN VOIDRAY> <BUILD PYLON> <RESEARCH AIRWEAPONS_LEVEL1> <TRAIN PROBE> x 3 <BUILD ASSIMILATOR> <TRAIN VOIDRAY> <TRAIN PROBE> x 2 <TRAIN VOIDRAY> <TRAIN PROBE> <BUILD PYLON> <TRAIN PROBE> x 4 <TRAIN ADEPT> <TRAIN VOIDRAY> <TRAIN PROBE> <BUILD ROBOTICSFACILITY> <TRAIN VOIDRAY> <TRAIN PROBE> x 5 <BUILD FLEETBEACON> <TRAIN PROBE> <BUILD STARGATE> <TRAIN PROBE> <BUILD NEXUS> <BUILD PYLON> x 2 <TRAIN PROBE> x 3 <BUILD SHIELDBATTERY> <TRAIN PROBE> x 3 <BUILD PYLON> <TRAIN CARRIER> x 2 <TRAIN PROBE> x 4 <BUILD ASSIMILATOR> <TRAIN ZEALOT> <BUILD ASSIMILATOR> <TRAIN PROBE> x 2

advisor B suggested
**Immediate Steps**: Focus on saturating your economy by training the queued Probes to reach optimal mineral and gas accumulation, ensuring you can support a larger army.
**Short-Term Actions**: Prioritize building additional Pylons to avoid supply block and maintain unit production, while also deploying an Orthotomist for map control and harassment.
**Long-Term Strategy**: Transition into a strong air force by continuously producing Carriers from your Fleet Beacon, while upgrading your air weapons to enhance your unit effectiveness for mid-game engagements.

So my advice is <TRAIN PROBE> x 3 <TRAIN ADEPT> x 2 <RESEARCH VOIDRAY_SPEED_UPGRADE> <BUILD PYLON> <TRAIN PROBE> <BUILD SHIELDBATTERY> <TRAIN PROBE> <BUILD PYLON> <TRAIN CARRIER> <TRAIN PROBE> x 2 <BUILD PYLON> x 2 <BUILD NEXUS> <BUILD PYLON> <TRAIN CARRIER> <BUILD ASSIMILATOR> x 2 <TRAIN CARRIER> <BUILD PHOTONCANNON> <BUILD SHIELDBATTERY> <TRAIN PROBE> x 3 <TRAIN CARRIER> <BUILD STARGATE> <TRAIN CARRIER> <BUILD PHOTONCANNON> <TRAIN CARRIER> <BUILD PYLON> <RESEARCH AIRWEAPONS_LEVEL1> <TRAIN PROBE> x 2 <TRAIN VOIDRAY> <TRAIN PROBE> x 2 <BUILD PHOTONCANNON> <TRAIN PROBE> <BUILD PYLON> <TRAIN PROBE> x 3 <BUILD ROBOTICSFACILITY> <TRAIN PROBE> x 2 <BUILD PHOTONCANNON> <TRAIN PROBE> <BUILD PYLON>

and advisor C suggested
**Immediate Steps**: Focus on saturating your mineral lines by training all available Probes to maximize your economy; aim for 80+ mineral accumulation.
**Short-term Actions**: In preparation for a potential push, ensure you have a healthy mix of units by warping in additional Zealots and Stalkers while also planning for a Rapid fire upgrade.
**Long-term Strategy**: Transition into a balanced army composition with more Gateway units and tech up by researching upgrades at the Forge, while expanding to a fourth Nexus to secure your economy long-term.

So my advice is <TRAIN PROBE> <TRAIN VOIDRAY> <BUILD GATEWAY> <TRAIN PROBE> x 2 <BUILD GATEWAY> <TRAIN PROBE> x 2 <BUILD ASSIMILATOR> x 2 <TRAIN PROBE> x 3 <TRAIN ZEALOT> x 2 <TRAIN PROBE> <TRAIN VOIDRAY> <TRAIN OBSERVER> <BUILD NEXUS> <TRAIN PROBE> x 2 <BUILD PYLON> <TRAIN PROBE> <TRAIN VOIDRAY> <TRAIN OBSERVER> <BUILD PYLON> x 2 <TRAIN PROBE> <BUILD PYLON> <TRAIN PROBE> x 4 <BUILD CYBERNETICSCORE> <TRAIN PROBE> x 2 <BUILD PYLON> <TRAIN PROBE> <TRAIN STALKER> x 3 <TRAIN PROBE> x 2 <BUILD GATEWAY> x 2 <BUILD PYLON> <TRAIN PROBE> <RESEARCH GROUNDWEAPONS_LEVEL1> <TRAIN VOIDRAY> <TRAIN PROBE> x 2 <BUILD PYLON> <TRAIN PROBE> <TRAIN VOIDRAY> <BUILD PYLON> <TRAIN PROBE>

We are following Advisor A`s opinion.
IMPORTANT: YOUR PREVIOUS RESPONSE <TRAIN CARRIER> IS FAILED BECAUSE FLEETBEACON IS NOT_EXIST!!!

---

Figure 12: **Input example for the Protoss race.** The imitation agent's input consists solely of the current game state. The strategic planner's input includes not only the current game state but also the imitation agent's output and feedback from the environment.

---

**Imitation Agent`s Output**

**Immediate Steps**: Focus on stabilizing your economy by training more Probes (up to 60) to ensure a strong resource income for future unit production.
**Short-Term Actions**: Transition into a carrier-heavy composition by continuously producing Voidrays and Carriers while also building additional Pylons to avoid supply block.
**Long-Term Strategy**: Aim for a robust air force with 5-7 Carriers supported by Adepts and Sentries to counter ground threats while expanding your economy with more Nexuses and Assimilators.

So my advice is <TRAIN PROBE> x 4 <BUILD PYLON> <TRAIN PROBE> <TRAIN VOIDRAY> <BUILD ASSIMILATOR> <TRAIN PROBE> <TRAIN VOIDRAY> <BUILD PYLON> <RESEARCH AIRWEAPONS_LEVEL1> <TRAIN PROBE> x 3 <BUILD ASSIMILATOR> <TRAIN VOIDRAY> <TRAIN PROBE> x 2 <TRAIN VOIDRAY> <TRAIN PROBE> <BUILD PYLON> <TRAIN PROBE> x 4 <TRAIN ADEPT> <TRAIN VOIDRAY> <TRAIN PROBE> <BUILD ROBOTICSFACILITY> <TRAIN VOIDRAY> <TRAIN PROBE> x 5 <BUILD FLEETBEACON> <TRAIN PROBE> <BUILD STARGATE> <TRAIN PROBE> <BUILD NEXUS> <BUILD PYLON> x 2 <TRAIN PROBE> x 3 <BUILD SHIELDBATTERY> <TRAIN PROBE> x 3 <BUILD PYLON> <TRAIN CARRIER> x 2 <TRAIN PROBE> x 4 <BUILD ASSIMILATOR> <TRAIN ZEALOT> <BUILD ASSIMILATOR> <TRAIN PROBE> x 2

---

**Strategic Planner`s Output**

```
<BUILD FLEETBEACON>
<TRAIN PROBE> x 4
<BUILD PYLON>
<TRAIN PROBE>
<TRAIN VOIDRAY>
<BUILD ASSIMILATOR>
<TRAIN PROBE>
<TRAIN VOIDRAY>
<BUILD PYLON>
<RESEARCH AIRWEAPONS_LEVEL1>
<TRAIN PROBE> x 3
<BUILD ASSIMILATOR>
<TRAIN VOIDRAY>
<TRAIN PROBE> x 2
<TRAIN VOIDRAY>
<TRAIN PROBE>
<BUILD PYLON>
<TRAIN PROBE> x 4
<TRAIN ADEPT>
<TRAIN VOIDRAY>
<TRAIN PROBE>
<BUILD ROBOTICSFACILITY>
<TRAIN VOIDRAY>
<TRAIN PROBE> x 5
<BUILD FLEETBEACON>
<TRAIN PROBE>
<BUILD STARGATE>
<TRAIN PROBE>
<BUILD NEXUS>
<BUILD PYLON> x 2
<TRAIN PROBE> x 3
<BUILD SHIELDBATTERY>
<TRAIN PROBE> x 3
<BUILD PYLON>
<TRAIN CARRIER> x 2
<TRAIN PROBE> x 4
<BUILD ASSIMILATOR>
<TRAIN ZEALOT>
<BUILD ASSIMILATOR>
<TRAIN PROBE> x 2
```

Figure 13: **Output example for the Protoss race.** We demonstrate that the strategic planner's output currently includes only its decisions, but by modifying the system prompt, we can incorporate the rationale into the output as well.

## O   Qualitative Results of the Feedback System

We illustrate how the HIMA architecture responds in two representative feedback scenarios. Figure 14 shows an infeasible action being received, while Figure 15 demonstrates coming under attack.

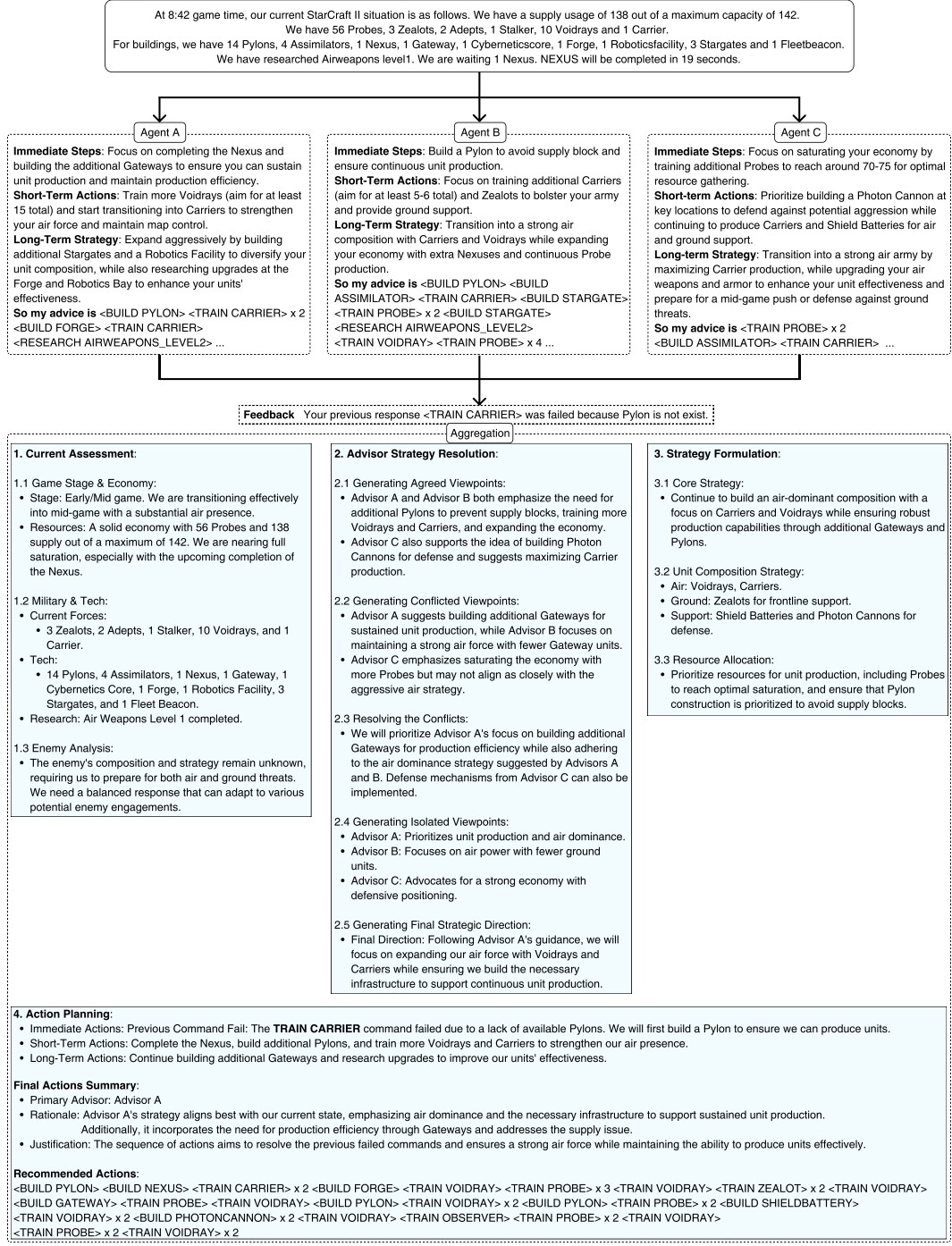

Figure 14: **Overall Strategy Generation Pipeline in an Infeasible Action Scenario.** Strategy planner (SP) arranges the action sequence so that the prerequisites for an previously infeasible action are fulfilled, allowing the action to become feasible and eventually be executed.

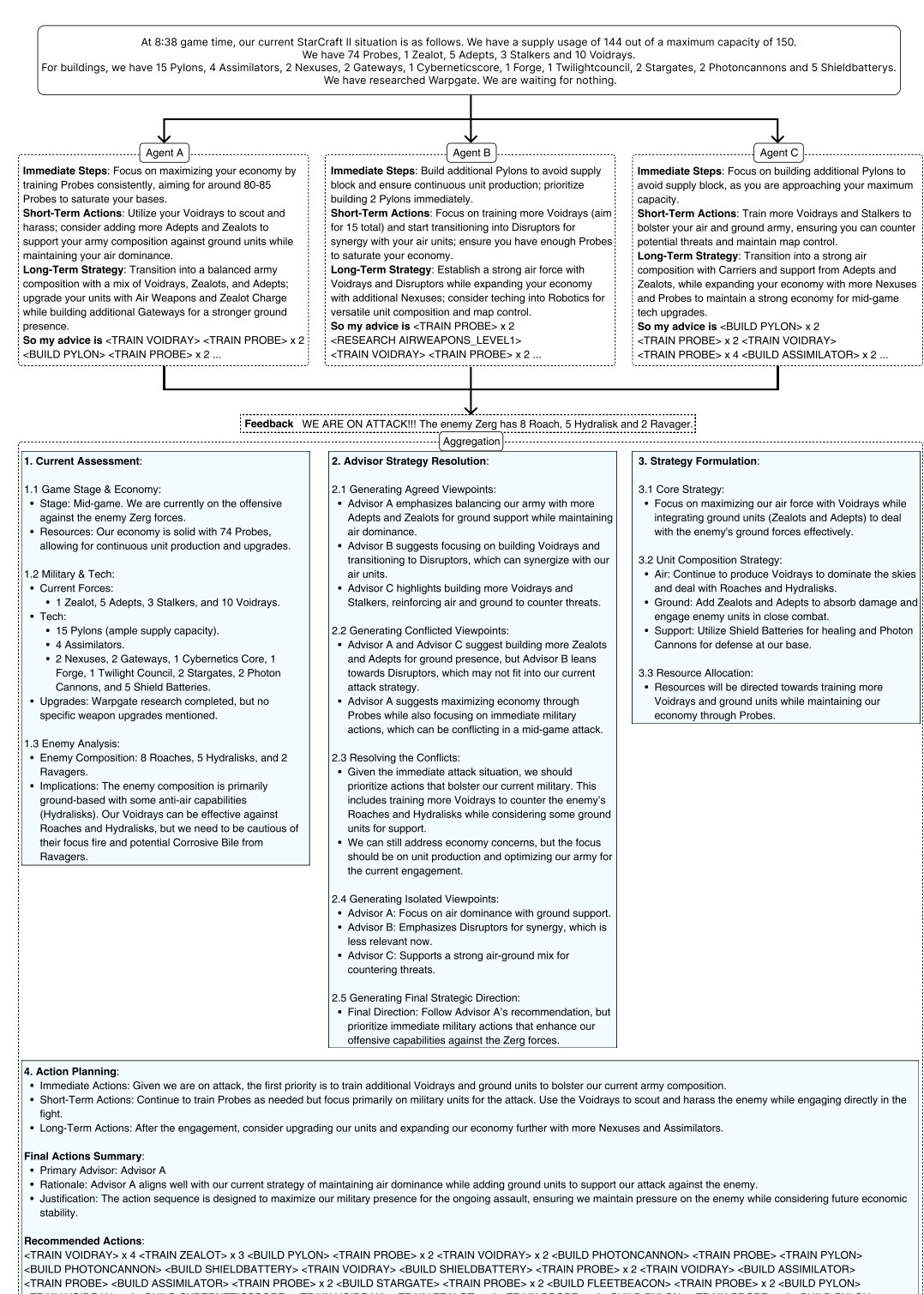

Figure 15: **Overall Strategy Generation Pipeline in a Defense Scenario.** Strategy planner (SP) urgently produces counter units (Void Rays, Zealots) in response to the enemy's attack, and also constructs defensive buildings (Photon Cannons) to prepare for subsequent assaults.

