# OpenReview forum: "Society of Mind Meets Real-Time Strategy: A Hierarchical Multi-Agent Framework for Strategic Reasoning"
_colmweb.org/COLM/2025/Conference — COLM 2025_

### Official Review · Reviewer_W96F · 2025-05-12

**Rating:** 6
**Confidence:** 4
**Ethics Flag:** 1

**Summary:**

This paper introduces a hierarchical imitation multi-agent (HIMA) framework for StarCraft II (SC2), where specialized agents create structured action sequences, and a meta-controller integrates them into cohesive strategies. By leveraging human replay data with tactical rationales, their approach maintains build orders, minimizes LLM queries, and adapts effectively to changing conditions. Additionally, this paper also presents the TEXTSCII-ALL environment, encompassing all nine race matchups, as a comprehensive testbed for real-time strategy (RTS) games. Experimental results demonstrate that HIMA outperforms state-of-the-art (SoTA) baselines in both win rates and efficiency, highlighting the effectiveness of combining imitation-driven specialization with high-level coordination in complex real-time strategy scenarios.

**Questions To Authors:**

For all my major concerns and questions, please see the Reasons To Reject section above. One suggestion is that the author should mention the experiment trial number in the main context and figure caption to help the readers assess the robustness and reliability of this proposed framework. Another suggestion is that it would be better to include human-human performance in this paper to highlight the upper capability boundary of this environment will increase the contribution and the significance of this proposed framework (comparison of human-human corporation and proposed framework).

**Reasons To Accept:**

1. This paper introduces a efficient framework (i.e., the HIMA framework ) that deploys specialized imitation agents trained on human demonstration data for generating diverse structured action sequences, coordinated by a Strategic Planner (SP), to address the limitations of existing LLM-based methods, such as producing short-horizon actions and invalid build orders.

2. This paper implements experiments to demonstrate that the proposed framework surpasses built-in AI opponents and direct match-ups with SoTA methods.

3 . The paper describes the framework and modules clearly and makes it easy to follow and understand. Generally, most of the paper is well written except for some figure captions and details.

**Reasons To Reject:**

Although I appreciate the effort and interesting framework in this paper, some weaknesses need to be improved.

1. The number of trials for each experiment is not mentioned in section 4.1 Experimental Setup. It is important to determine the effectiveness and reliability of the proposed method. Please mention the trial number in the Experimental Setup section and the caption of each experimental figure and table.

 2. Since the main contribution of this paper is the proposed framework methods, it lacks of comparison with the reinforcement learning baseline or the large language models baseline (e.g., o1 or deepseek r1). Moreover, it is better to add the human performance as the upper bound because the proposed method uses imitation learning, which uses human demonstrations to capture the core patterns of expert play.

3. The proposed framework only employs the Qwen-2 1.5B as the imitation learning agent base. To demonstrate the effectiveness of this framework, it should adopt at least one more open source model (e.g., llama) as the imitation learning agent base. Furthermore, comparing the closed-source model (e.g., gpt4o) as a specific agent to show the contribution of the imitation learning method.

4. Since the second contribution of this paper is expanding the SC2 evaluation environment,  what is the difference between the proposed evaluation environment and the SC2 evaluation environment? The difference between these two environments is the action space mentioned in the appendix (e.g., the Protoss have 58 possible actions, 475 the Zerg 61, and the Terrans 62)? The detailed comparison should be the main context of the paper.

---

> ### Author Response · Authors · 2025-06-01
>
> ## Response to Reviewer W96F (1/2)
>
> We thank Reviewer **W96F** for the constructive feedback and positive comments on our HIMA framework, experimental results, and clear presentation. For additional experiments, we focus on the Protoss (player) vs. Zerg (opponent) matchup, the most commonly studied scenario in prior work [1,2,3]. We hope our response addresses your concerns.
>
> > The number of trials for each experiment is not mentioned. Please specify the trial number.
>
> $\to$ As noted, the number of trials is already specified in the manuscript (L207, L228, L233): 50 randomly sampled matches for the main evaluation (Section 4.1, L207), 10 consecutive games for HIMA vs. baselines (L228), and 20 randomly sampled matches for detailed analysis (Section 4.4, L233). For clarity, we will also state the number of trials in all relevant table and figure captions in the revised manuscript. Thank you for your helpful feedback!
>
> > Including human evaluation in this paper to highlight the upper capability boundary of this environment will increase the contribution and the significance of this proposed framework.
>
> $\to$ Good suggestion! For an upper boundary, we first conduct a human evaluation for human vs. built-in AI in the Protoss (human) vs. Zerg (AI) matchup, and their win rates are summarized in the table below. Experimental results with various human players (novice, gold, and platinum) show that platinum and gold-level players consistently outperform the built-in AI, setting an upper bound for agent performance. Here, MMR refers to the matchmaking rating used in StarCraft II to measure a human player's skill level.
>
> | Human Player | Rank     | MMR   | Difficulty Lv.5   | Difficulty Lv.6   | Difficulty Lv.7   |
> |--------------|----------|-------|-------------------|-------------------|-------------------|
> | Player A     | Platinum | 3,100 | 100% (5/5)        | 100% (5/5)        | 100% (5/5)        |
> | Player B     | Gold     | 2,610 | 100% (5/5)        | 80% (4/5)         | 80% (4/5)         |
> | Player C     | Novice   | -     | 40% (2/5)         | 0% (0/5)          | 0% (0/5)          |
>
> We also evaluate HIMA against human players in the Protoss (HIMA) vs. Zerg (human) matchup. HIMA consistently outperforms novice and gold-level players, and performs on par with platinum-level players, demonstrating the practical effectiveness and significance of our framework.
>
> | Human Player | Rank     | MMR   | Win-rate of HIMA   |
> |--------------|----------|-------|--------------------|
> | Player A     | Platinum | 3,100 | 40% (2/5)          |
> | Player B     | Gold     | 2,610 | 80% (4/5)          |
> | Player C     | Novice   | -     | 100% (5/5)         |

---

> > ### Author Response · Authors · 2025-06-01
> >
> > ## Response to Reviewer W96F (2/2)
> >
> > > Adopt at least one more open source model (e.g., Llama) beyond Qwen-2 1.5B as the imitation learning agent.
> >
> > $\to$ Thank you for your suggestion! We conduct additional experiments with various small agent models beyond Qwen-2 1.5B, and the results are summarized in the table below. These results show that all tested models achieve similar performance in our setting, even though their performance differences may be more noticeable on other benchmarks. We believe this is because all models are trained on the same imitation learning data, which minimizes performance gaps between them.
> >
> > | Model      | Imitation agent size | Win-rate (%) at Lv.7  |
> > |------------|---------------------|-----------------------|
> > | Qwen-2     | 1.5B x 3            | 82 (41/50)            |
> > | Qwen-3     | 1.7B x 3            | 85 (17/20)            |
> > | Llama-3.2  | 3B x 3               | 85 (17/20)            |
> >
> >
> > > Comparing the closed-source model (e.g., gpt4o) as a specific agent to show the contribution of the imitation learning method.
> >
> > $\to$ Great suggestion! We compare HIMA with a multi-agent setup using the closed-source model (gpt-4o), which was implemented by making three independent predictions, as shown in the table below. Performance drops significantly compared to our imitation multi-agent approach (HIMA), which we attribute to the effectiveness of imitation learning agents trained on human demonstrations in HIMA. Thank you for your valuable feedback!
> >
> > | Agent Configuration                                          | Win-rate (%) at Lv.7 |
> > |--------------------------------------------------------------|----------------------|
> > | Closed-source model (GPT-4o) — with a strategic planner      | 20 (2/10)            |
> > | Imitation multi-agent — with a strategic planner **(Ours)**       | 82 (41/50)           |
> >
> >
> > > What is the difference between the proposed evaluation environment and the SC2 evaluation environment? Is it the action space as mentioned in the appendix?
> >
> > $\to$ Partially, yes. The main difference is not only the expanded action space, but also the broader scope of evaluation. Our proposed TEXTSCII-ALL supports all three races and all nine possible matchups, whereas the previous SC2 environment only allowed Protoss (player) vs. Zerg (opponent). To support this, we define and implement full action spaces for Terran and Zerg, enabling any race to be played as either player or opponent. This makes comprehensive and fair benchmarking possible. We will include discussions of these differences between the proposed evaluation environment and the SC2 environments in the revised manuscript. Moreover, all code for the new evaluation environment will be made publicly available.
> >
> > [1] Li et al., Hierarchical Expert Prompt for Large-Language-Model …, arXiv, 2025.\
> > [2] Ma et al., Large language models play starcraft ii …, NerIPS, 2024.\
> > [3] Wu et a., LLMs Are Not Good Strategists …, ICLR workshop, 2025.

---

> > > ### Author Response · Authors · 2025-06-07
> > >
> > > We sincerely thank you for your effort in reviewing our submission. We gently remind the reviewer that we tried our best to address your concerns via our replies. As the discussion period is nearing the end, we would be delighted to hear more from you if there are any further concerns.

---

> > ### Comment · Reviewer_W96F · 2025-06-09
> > **Official Comment by Reviewer W96F**
> >
> > Thank you for your response to the comments. I appreciate the effort you put into addressing the issues raised. Since the author has addressed all my concerns, I will raise my score.

---

### Official Review · Reviewer_crGH · 2025-05-12

**Rating:** 6
**Confidence:** 4
**Ethics Flag:** 1

**Summary:**

This paper proposes a hierarchical multi-agent framework to enable more general and adaptable embodied intelligence in robotic systems. The authors integrate multiple specialized sub-agents that operate in parallel and are coordinated. By imitation learning (actually is SFT) different LLM agent with different roles, they demonstrate the feasibility of this approach using a virtual embodied agent.

**Questions To Authors:**

The work only use one base model, i.e.,  Qwen-2 1.5B, in the evaluation. It is suggested to include more base models.

**Reasons To Accept:**

1), The topic is interesting. Basically, using LLM agets for emboded AI is interesting research area. The work is focusing on StarCraft II.

2). The work builds the pipeline with imitated LLM agents (SFT) different roles is proved an effective way.

3). Experiments validate the effectiveness of the proposed approach.

**Reasons To Reject:**

1.It is unclear how well this architecture would scale outside the simulation environment or handle more complex physical tasks.

2. The paper is not well-formulated and very hard to follow. Many terms are used without well definition.

3. The idea basically is not very novel. Actually, the idea is not exact imitation, actually is role-wise fine-tuning (SFT), which is not new in the literature.

4. The work only use one base model, i.e.,  Qwen-2 1.5B, in the evaluation. It is suggested to include more base models.

---

> ### Author Response · Authors · 2025-06-01
>
> ## Response to Reviewer crGH
>
> We thank Reviewer **crGH** for the constructive feedback and positive comments on our use of LLM agents for embodied AI in StarCraft II, the effectiveness of our pipeline, and the experimental validation of our approach. For additional experiments, we focus on the Protoss (player) vs. Zerg (opponent) matchup at difficulty level 7, the most commonly studied scenario in previous work [1,2,3]. We hope our response addresses your concerns.
>
> > It is unclear how well this architecture would scale outside the simulation environment or handle more complex physical tasks.
>
> $\to$ HIMA is designed to scale beyond simulation and handle complex real-world tasks, thanks to its hierarchical and modular architecture. Although our current experiments focus on StarCraft II, the core principles underlying HIMA—such as specialized agent collaboration, imitation-based learning, and adaptive planning—directly address challenges common in physical environments, including partial observability and dynamic changes. These strengths make HIMA well-suited for applications like multi-robot coordination and physical system control. We discuss scalability and real-world applicability further in the revised manuscript.
>
> > The paper is not well-formulated and very hard to follow. Many terms are used without well definition.
>
> $\to$ We appreciate your feedback and see the review process as a chance to clarify any unclear aspects of the paper. If you can specify which terms or sections are hard to follow, we will gladly address them in the review process. Your detailed input will help us make the paper clearer and more accessible.
>
> > The idea basically is not very novel. Actually, the idea is not an exact imitation, actually is role-wise fine-tuning (SFT), which is not new in the literature.
>
> $\to$ The novelty of our work lies not in the basic use of imitation learning or role-wise fine-tuning (SFT), but in our hierarchical, multi-agent structure and meta-level orchestration using a Strategic Planner (SP). While standard imitation learning and SFT struggle to adapt in novel or dynamic scenarios, our framework addresses this limitation by introducing a SP that coordinates multiple specialized imitation agents (L44 and L59, Sec. 3). This planner integrates agent outputs, incorporates environment-aware reasoning, and leverages temporal Chain-of-Thought planning for effective long-term planning. As shown in Tables 2 and 4 and Figure 4, this design enables robust adaptability and strategic coherence in complex environments, clearly distinguishing our method from prior work.
>
> > The work only uses one base model (Qwen-2 1.5B) in the evaluation. It is suggested to include more base models.
>
> $\to$ Thank you for your suggestion! We show results w.r.t various small agent models in the table below. Results show that all tested models achieve similar performance in our setting, even though their performance differences may be more noticeable on other benchmarks. We believe this is because all models are trained on the same imitation learning data, which minimizes performance gaps between them.
>
> | Model      | Imitation agent size | Win-rate (%) at Lv.7  |
> |------------|---------------------|-----------------------|
> | Qwen-2     | 1.5B x 3            | 82 (41/50)            |
> | Qwen-3     | 1.7B x 3            | 85 (17/20)            |
> | Llama-3.2  | 3B x 3               | 85 (17/20)            |
>
> [1] Li et al., Hierarchical Expert Prompt for Large-Language-Model …, arXiv, 2025.\
> [2] Ma et al., Large language models play starcraft ii …, NerIPS, 2024.\
> [3] Wu et a., LLMs Are Not Good Strategists …, ICLR workshop, 2025.\

---

> > ### Comment · Reviewer_crGH · 2025-06-03
> > **Thanks for the response**
> >
> > Although I still think the paper is not well-written and hard to follow, i would raise my score. Strongly suggest the authors to improve their writing in the new version.

---

> > > ### Author Response · Authors · 2025-06-03
> > >
> > > We sincerely appreciate your effort in reviewing our submission. We definitely will revise the manuscript as you recommended to improve clarity and readability. Your valuable feedback motivates us to enhance the writing in the new version, and we are committed to making the paper easier to follow.

---

> > > ### Author Response · Authors · 2025-06-07
> > >
> > > As the discussion period is coming to a close, we would like to gently remind you of your earlier mention of the possibility of raising the score. If there are any remaining points that require further clarification, please do not hesitate to let us know—we are more than willing to provide additional explanations. We sincerely appreciate your thoughtful feedback and the valuable opportunity to improve our work. Thank you again for your careful review.

---

### Official Review · Reviewer_RWbE · 2025-05-12

**Rating:** 5
**Confidence:** 4
**Ethics Flag:** 1

**Summary:**

The paper presents a new method for the text version of StarCraft II with LLMs. The authors introduce a hierarchical agent with a meta-controller (strategic planner) that produces mult-step actions. These actions are first proposed by multiple agents with different playing styles and then aggregated by the strategic planner. The authors also introduce SCII-All, which extends SC2 evaluations to all race conditions.

**Reasons To Accept:**

- Compared to previous baselines, the method seems to work well.
- The ablations are clear and make the role of certain parts of the method. I like the analyses of temporal cot, planning horizon vs win rates with api calls.
- Comprehensive benchmark expansion to nine race combinations.

**Reasons To Reject:**

- The paper is very unclear. A lot of the sections of the paper, in particular, the method, are difficult to understand. For example, terms like Nominal group technique, Strategic Objective aren't explained, and datasets central to the method like SC2EGSet aren’t explained well.
- The problem is not well motivated and the design decisions seem arbitrary at points.
- No error bars or significance tests.
- The method uses GPT-4o, a closed-source model. No open-source/open weights model is tested.
- In Table 4, with the strategic planner being GPT-4o, the configurations aren’t compute matched. This table felt a little misleading initially, as the model size column doesn’t include the size of the strategic planner.
- Only one model has been tested as the strategic planner? Does the method transfer to other models?
- The method seems very specific to improving performance on StarCraft. There is no discussion on why the techniques introduced are general or why they would work for domains beyond StarCraft.

---

> ### Author Response · Authors · 2025-06-01
>
> ## Response to Reviewer RWbE (1/2)
>
> We thank Reviewer **RWbE** for the constructive feedback and positive comments on our method’s strong performance, clear ablation studies, and comprehensive benchmark expansion. For additional experiments, we focus on the Protoss (player) vs. Zerg (opponent) matchup at difficulty Level 7, which is the most commonly studied scenario in prior work [3,4,5]. We hope our response addresses your concerns.
>
> > The method parts are difficult to understand, particularly the Nominal Group Technique, Strategic Objective, and SC2EGSet.
>
> $\to$ The Nominal Group Technique is a structured decision-making principle we apply to aggregate proposals from multiple agents, as described in [1] (L155) and shown in Figures 3-(b), 14, and 15. This process has five stages: generating agreed and conflicted viewpoints, resolving conflicts, generating isolated viewpoints, and formulating the final strategy (Figures 14 and 15). The Strategic Objective is the high-level goal assigned to each specialized agent based on expert gameplay analysis (L158), with examples in Appendix K and Figure 10. SC2EGSet, a dataset of professional StarCraft II replays with game states and actions [2], is used for training and evaluation, as detailed in Figure 3 (left) and Appendix E.1. We will clarify these in the revised version.
>
> > The problem is not well motivated and the design decisions seem arbitrary at points.
>
> $\to$ Our work is motivated by the challenges of real-time strategy games like StarCraft II, where existing LLM-based methods [3,4,5,6] struggle with strategic coherence and adaptability in dynamic, partially observable environments (see L37 and L44). To address these limitations, HIMA incorporates an imitation multi-agent component to support diverse and robust long-horizon planning, combined with a Strategic Planner that enables adaptive, environment-aware decision-making within a hierarchical structure.
>
> > No error bars or significance tests.
>
> $\to$ We conduct statistical tests to compare HIMA’s win rates with previous methods in Table 2. In Protoss (player) vs. Zerg (opponent) at difficulty 7, HIMA achieves an 82% win rate (50 trials, L207), which is statistically significant compared to HEP’s 25% (12 trials in [3], p ≈ 0.00015). At difficulty 6, HIMA’s 84% win rate (50 trials) is also statistically significant compared to TextStarCraft’s 8% (12 trials in [4], p-value ≈ 0.00000117) and EpicStar’s 30% (40 trials in [5], p-value ≈ 0.00000000344).
>
> In the Zerg (player) vs. Terran (opponent) matchup, the difference between SwarmBrain [6] and HIMA is not statistically significant, likely due to limited samples. Increasing HIMA’s trials to 150 yields an 84% win rate (±1.5 standard deviation) at difficulty 5; although not statistically significant, HIMA’s average remains higher than SwarmBrain’s 76%, given the observed variance.
>
> Across all detailed analyses (Tables 4 and 5, each with 20 repetitions), HIMA consistently shows statistically significant improvements (all p < 0.05).

---

> > ### Author Response · Authors · 2025-06-01
> >
> > ## Response to Reviewer RWbE (2/2)
> >
> > > The method uses GPT-4o, a closed-source model. No open-source/open weights model is tested. Only one model has been tested as the strategic planner? Does the method transfer to other models?
> >
> > $\to$ Yes, we used only one model (GPT-4o-mini) as the Strategic Planner (SP). However, other models can also be used, so we test various open- and closed-source models, as shown in the table below. Closed-source GPT-4o model achieves the best performance. Among open-source models, larger ones (e.g., Qwen-2.5 72B) also perform well and in some cases come close to matching the performance of closed-source models (e.g., GPT-4o-mini and Claude).
> > | Model         | Size  | Win-rate (%) at Lv.7    |
> > |---------------|-------|--------------------------|
> > | Qwen-2.5      | 32B   | 60 (12/20)               |
> > | Qwen-2.5      | 72B   | 75 (15/20)               |
> > | Qwen-3        | 8B    | 15 (3/20)                |
> > | Qwen-3        | 32B   | 55 (11/20)               |
> > | Llama-3.3     | 70B   | 65 (13/20)               |
> > | GPT-4o-mini   | -     | 82 (41/50)               |
> > | GPT-4o        | -     | 90 (18/20)               |
> > | Claude-sonnet | -     | 70 (14/20)               |
> > | Claude-haiku  | -     | 60 (12/20)               |
> >
> > > In Table 4, the configurations aren’t compute-matched since the model size column excludes the GPT-4o strategic planner, which may be misleading.
> >
> > $\to$ Sorry for the confusion. To avoid confusion, we will revise the column name to ‘Imitation agent size’ instead of ‘Model size’ in the updated manuscript. Thank you for pointing this out.
> >
> >
> > > The method seems specific to StarCraft, with no discussion of why the proposed techniques are general or effective beyond this domain.
> >
> > $\to$ While our experiments focus on StarCraft II, the techniques in HIMA are broadly applicable. Hierarchical coordination, imitation-based training, and environment-aware long-horizon planning are relevant to domains like robotics, autonomous driving, industrial automation, and multi-agent simulations. The feedback-driven strategic planner and temporal chain-of-thought reasoning can enhance adaptive decision-making in any dynamic environment. We will add discussion of these broader applications in the revised manuscript.
> >
> > [1] Long et al., Multi-expert prompting improves …, EMNLP, 2024.\
> > [2] Bialecki et al., SC2EGSet: StarCraft II Esport Replay…, Scientific Data, 2023.\
> > [3] Li et al., Hierarchical Expert Prompt for Large-Language-Model …, arXiv, 2025.\
> > [4] Ma et al., Large language models play starcraft ii …, NerIPS, 2024.\
> > [5] Wu et a., LLMs Are Not Good Strategists …, ICLR workshop, 2025.\
> > [6] Shao et al., SwarBrain: Embodied agent for real-time strategy …, arXiv, 2024.\

---

> > > ### Comment · Reviewer_RWbE · 2025-06-03
> > >
> > > Thank you for your response! The additional experiments with open-source weights models improve the work. I will raise my score by a point, but I am still not convinced of why the problem formulation is general, beyond starcraft. And the presentation of the paper is still not great.

---

> > > > ### Author Response · Authors · 2025-06-11
> > > >
> > > > As the discussion period is coming to a close, we would like to gently remind you that we are happy to provide further clarification regarding the generality of our approach or any other points that may need further explanation. If you have any additional questions or concerns, please feel free to let us know. Thank you again for your thoughtful feedback.

---

> > ### Author Response · Authors · 2025-06-04
> >
> > > I am still not convinced of why the problem formulation is general, beyond starcraft.
> >
> > $\to$ Thanks for your continued discussion! To clarify, our central problem is to enable coherent and adaptive long-term sequential decision-making in complex environments that are both dynamic and partially observable. We believe these challenges are not unique to StarCraft II; they frequently arise in many real-time strategy (RTS) simulations and real-world domains where systems must plan over long horizons, adapt to unpredictable changes, and operate with incomplete information.
> >
> > The generality of our approach stems from the design principles underlying HIMA. Our framework introduces a hierarchical structure that separates high-level, environment-aware strategic planning from specialized tactical execution, each focusing on particular objectives or roles. By leveraging structured action planning (Sec. 3.1) and temporal chain-of-thought aggregation with NGT (Sec. 3.2), HIMA can flexibly incorporate expert knowledge and adapt to novel scenarios. Furthermore, the feedback-driven planner further ensures that the system can correct or update its strategy in real-time when faced with new or evolving situations (L174). Importantly, HIMA’s hierarchical and modular design allows for the creation and integration of new tactical agents tailored to different domains, as long as suitable demonstration data is available. This makes the framework readily adaptable to a wide range of simulation and real-world applications beyond StarCraft II, as also noted in reviewer bnMe’s comments regarding Technological Impact and Understanding Depth.
> >
> > While our empirical validation is anchored in the StarCraft II domain due to its complexity and benchmarking value, the architectural principles and algorithms are designed to be domain-agnostic. We believe that HIMA can be applied to any setting that requires robust, adaptive, and scalable long-term decision-making under uncertainty, and we plan to elaborate on these broader implications in our revised manuscript.
> >
> > > The presentation of the paper is still not great.
> >
> > $\to$ We appreciate your feedback and view the review process as an opportunity to clarify and improve our work. If you could specify which sections or terms you found unclear or difficult to follow, we would be happy to address them in the review process. Your detailed input will be invaluable in helping us make the paper clearer and more accessible.

---

> > ### Author Response · Authors · 2025-06-07
> >
> > As the discussion period is coming to a close, we would like to gently remind you that we are happy to provide further clarification regarding the generality of our approach or any other points that may need further explanation. If you have any additional questions or concerns, please feel free to let us know. Thank you again for your thoughtful feedback.

---

### Official Review · Reviewer_bnMe · 2025-05-13

**Rating:** 7
**Confidence:** 4
**Ethics Flag:** 1

**Summary:**

This paper proposes a hierarchical multi-agent framework (HIMA) for strategic reasoning in real-time strategy games, specifically focusing on StarCraft II. The framework utilizes specialized agents and a meta-controller to achieve state-of-the-art performance compared to LLM-based approaches. Additionally, the paper introduces the TEXTSCII-ALL environment for evaluating performance across all race matchups in StarCraft II.

**Reasons To Accept:**

**Empiricism, Data, and Evaluation**: The paper demonstrates that HIMA achieves superior scores compared to other LLM-based approaches, providing a strong empirical foundation. The introduction of the TEXTSCII-ALL benchmark is a significant contribution, as it enhances the evaluation of performance across various race matchups. Furthermore, the incorporation of tactical rationale into existing datasets using LLMs is a novel contribution that enhances the instruct-tuned dataset with medium- to long-term goals.

**Technological Impact and Understanding Depth**: By allowing agents to output action sequences, the framework promotes more consistent behavior, which is crucial for effective decision-making in complex environments. The use of clustering to analyze data and imbue agents with specialization is a noteworthy aspect of the work, and this automation could be applicable to other problems, enhancing its relevance. This ambitious approach to tackling complex long-term decision-making with LLMs highlights the potential for significant advancements in the field.

**Efficiency**: The reduction in the number of LLM calls due to long-term planning makes this approach applicable to real-time scenarios, which is a significant advantage.

**Reasons To Reject:**

While the paper presents several strengths, it is limited to a single benchmark (StarCraft II), raising questions about the generalizability of the findings. Future work should explore the application of HIMA in other real-time strategy games to validate its effectiveness across different contexts.

---

> ### Author Response · Authors · 2025-06-01
>
> ## Response to Reviewer bnMe
>
> We thank Reviewer **bnMe** for the constructive feedback and positive comments on our method’s strong empirical results, the introduction of the TEXTSCII-ALL benchmark, and our novel use of LLMs to enhance datasets with tactical rationale. We also appreciate your recognition of our framework’s ability to output action sequences, leverage clustering for agent specialization, and improve efficiency through reduced LLM calls. We hope our response addresses your comments and concerns.
>
> > This paper is limited to a single benchmark (StarCraft II), raising questions about the generalizability of the findings. Future work should explore the application of HIMA in other real-time strategy games to validate its effectiveness across different contexts.
>
> $\to$ Thank you for your feedback. We recognize that our experiments are limited to StarCraft II and agree that testing HIMA in other environments is important for demonstrating its generalizability. As future work, we plan to apply HIMA to additional RTS environments such as Civilization [1], poker games [2], Minecraft [3] to evaluate its generalizability. We believe that HIMA’s hierarchical coordination, imitation-based learning, and environment-aware planning are broadly applicable to complex, sequential decision-making tasks, not only in games but also in domains like multi-agent robotics and industrial automation.
>
> [1] Allen et al., "Artificial Intelligence and Civilization: Using Deep Reinforcement Learning to Play Civilization VI," AAAI Workshop on Artificial Intelligence for Strategy Games, 2021.\
> [2] Zhuang et a., "PokerBench: Training Large Language Models to Become Professional Poker Players," AAAI, 2025.\
> [3] Guss et al., "The MineRL Competition on Sample Efficient Reinforcement Learning using Human Priors," NeurIPS Competition Track, 2019.

---

> > ### Author Response · Authors · 2025-06-07
> >
> > We sincerely thank you for your effort in reviewing our submission. We gently remind the reviewer that we tried our best to address your concerns via our replies. As the discussion period is nearing the end, we would be delighted to hear more from you if there are any further concerns.

---

> > > ### Comment · Reviewer_bnMe · 2025-06-11
> > >
> > > Thank you for the response and for outlining your plan to evaluate HIMA on the additional benchmarks. I think the new experiments should further clarify the method’s generalizability. While I look forward to seeing those results in a future revision or follow-up work, my overall assessment and score remain unchanged at this time.

---

### Decision · Program_Chairs · 2025-07-08

**Decision:**

Accept

**Comment:**

This paper presents a hierarchical framework for strategic reasoning for real-time games such as StarCraft II.  A key idea is to represent planning in text form, and utilize the text-based SCII API.

The reviewers are generally positive about this work.  One reviewer mentioned several areas where clarity could be improved, and the authors should revise the paper in those aspects.